# OUT-OF-DISTRIBUTION IMAGE DETECTION
# USING THE NORMALIZED COMPRESSION DISTANCE

## ABSTRACT

To reliably detect the *out-of-distribution* images based on already deployed convolutional neural networks, several recent studies on out-of-distribution detection have tried to define effective confidence scores without retraining the model. Although they have shown promising results, most of them need to find the optimal hyperparameter values by using a few out-of-distribution images, which eventually assumes a specific test distribution and makes it less practical for real-world applications. In this work, we propose a novel out-of-distribution detection method MALCOM, which neither uses any out-of-distribution samples nor retrain the model. Inspired by the observation that global average pooling cannot capture the spatial information of the feature maps from convolutional neural networks, our method aims to extract informative sequential patterns from the feature maps. To this end, we introduce a similarity metric which focuses on the shared patterns between two sequences based on normalized compression distance. In short, MALCOM uses both the global average and spatial pattern of the feature maps to accurately identify out-of-distribution samples.

## 1 INTRODUCTION

The *distributional uncertainty* refers to an uncertainty originated from the inconsistency between training and test distributions (Malinin & Gales, 2018). Recently, measuring the distributional uncertainty of deep neural networks has gained much attention, in order to detect the *out-of-distribution* sample which comes from outside the training distribution. In contrast to the existing belief that the softmax output of deep neural networks is not appropriate measure for uncertainty, Hendrycks & Gimpel (2017) found out that the softmax score is able to distinguish between in-distribution and out-of-distribution samples to a degree.

Since then, several methods have shown promising results for out-of-distribution detection (Liang et al., 2018; Lee et al., 2018b; Lakshminarayanan et al., 2017; Lee et al., 2018a), but the most of them still have limitations from a practical perspective in that they need to find the optimal values of hyperparameters using the out-of-distribution samples or retrain the model using a new objective function specifically designed for detection. For example, Liang et al. (2018) improved the ability of distinguishing the out-of-distribution data by using temperature scaling and input perturbation, which involves in several hyperparameters; this kind of calibration techniques require out-of-distribution samples for validation to adjust its hyperparameters to manipulate the softmax outputs. On the other hand, Lee et al. (2018a) defined an another loss function by adding the KL divergence, assuming that the softmax output of the out-of-distribution is uniform; it has to retrain the model because of the newly designed loss function. For these reasons, they have been difficult to be deployed for real-world applications. In this sense, our motivation is how far out-of-distribution detection can be done without using the out-of-distribution set for validation while employing the existing softmax classifier without retraining it.

The state-of-the-art method (Lee et al., 2018b) tries to detect out-of-distribution samples based on the Mahalanobis distance from class means instead of using the softmax output, by utilizing the feature maps obtained by convolutional neural networks. In particular, they estimate the class-conditional probabilities for each class, which are more appropriate and effective to detect out-of-distibution samples than the softmax output which provides a mere relative values between classes. However, they obtain the feature vectors of input images by averaging each feature map, also known as global

average pooling. We notice that the spatial information about input images disappears in the process of averaging feature vectors, and this eventually degrades the detection performance.

To tackle this challenge, we offer to use the *normalized compression distance* (NCD), which can measure the distance between two images without the loss of such spatial information of the feature maps. NCD is a general method of measuring the intrinsic distance between two arbitrary objects using the off-the-shelf compression algorithm (e.g. 7-zip, bzip, or lzw). With its solid theoretical background and general applicability, NCD is widely used in various fields, including brain diagnosis or spam filtering (Berek et al., 2014; Spracklin & Saxton, 2007). The high-level idea of NCD is to evaluate how many shared patterns are eliminated when two binary sequences are compressed together. By using the NCD, we aim to accurately measure how similar the feature maps of a test sample is with those of training examples, and use it to determine whether the test sample is from out-of-distribution or not. Our experiments demonstrate that the proposed method, termed as MAL-COM, achieves better or comparable performances for detecting out-of-distribution images to the existing methods.

## 2 BACKGROUNDS

### 2.1 CONSTRAINED OUT-OF-DISTRIBUTION DETECTION

Before proceeding to describe the notions closely related to our method, let us state what we want to tackle exactly: *out-of-distribution detection without retraining network and without sniffing out-of-distribution samples*. Starting with the definition, let $\mathcal{D}_{train} = \{(\boldsymbol{x}_i, y_i)\}_{i=1}^{N}$ be the training dataset and $\boldsymbol{f}(\cdot; \boldsymbol{\theta})$ be the model trained on $\mathcal{D}_{train}$. Suppose that each data $\boldsymbol{x}_i$ from $\mathcal{D}_{train}$ is an observation of random variable $\boldsymbol{X}_{\text{train}}$. Thereby, the out-of-distribution detection task is to obtain the out-of-distribution score function $\kappa(\cdot; \mathcal{D}_{train}, \boldsymbol{\theta}, \boldsymbol{\phi})$ for a discrimination between in-distribution and out-of-distribution. In other words, given a test instance $\boldsymbol{x}^*$, the score $\kappa(\boldsymbol{x}^*)$ should be high if $\boldsymbol{x}^*$ is not an observation of $\boldsymbol{X}_{\text{train}}$, otherwise should be low. There are many ways to get score function $\kappa$, but we work on the task under the following constraints:

- The score function parameter $\boldsymbol{\phi}$ should be independent with the distribution of test data $\boldsymbol{x}^*$, i.e., $\boldsymbol{\phi}$ is determined by only in-distribution. This is intuitive in that $\boldsymbol{x}^*$ does not assume any test distribution. This requirement implies that hyperparameter tuning by using out-of-distribution samples is not allowed.

- The model parameter $\boldsymbol{\theta}$ should not be changed, i.e., the trained model primitively for the purpose of classification should not be retrained. Then, the score function can be utilized by the neural networks which are already trained and deployed in real-world applications.

The existing methods that satisfy these conditions are the softmax output detector (Hendrycks & Gimpel, 2017) and the simplified Mahalanobis detector (Lee et al., 2018b).

### 2.2 MAHALANOBIS DETECTOR

The Mahalanobis detector (MAHALANOBIS), one of the best performing methods, introduces an effective method to remove the correlation between feature maps of the last hidden layer (Lee et al., 2018b), but combining layers leads to the degradation of the detection performance.

#### 2.2.1 MAHALANOBIS-VANILLA

We factor out a simplified version of MAHALANOBIS (MAHALANOBIS-vanilla), because the whole MAHALANOBIS detector needs out-of-distribution samples to tune the hyperparameters of input preprocessing and layer weights, which fails to satisfy our requirements (Sec 2.1). For the whole method of MAHALANOBIS which utilizes out-of-distribution samples to boost the performance, we refer the reader to see (Lee et al., 2018b).

MAHALANOBIS-vanilla uses the feature maps of the last hidden layer of convolutional neural networks. To be exact, we denote the output of the $l$-th hidden layer of the network $\boldsymbol{f}$ by $\boldsymbol{f}_l$ for $l = 1, \ldots, L$, where $L$ is the number of hidden layers. For the $l$-th hidden layer, let $\boldsymbol{f}_l^i$ be the $i$-th feature map of its output for $i = 1, \ldots, M_l$, where $M_l$ is the number of feature maps of the $l$-th

Table 1: Performance of different Mahalanobis setting for DenseNet. 'R' is the abbreviation for 'resized'. The best and worst results are highlighted in boldface and underline for each setting.

| ID | OOD | TNR at TPR 95% | AUROC | Detection Accuracy | AUPR(In) | AUPR(Out) |
|---|---|---|---|---|---|---|
| | | MAHALANOBIS-vanilla / MAHALANOBIS-assemble / MALCOM (ours) | | | | |
| CIFAR-10 | SVHN | 89.2 / 69.4 / **93.4** | 97.6 / 85.4 / **98.4** | 92.4 / 82.8 / **94.3** | 94.5 / 55.9 / **95.9** | 99.02 / 94.9 / **99.3** |
| | TinyIm(R) | 82.3 / 84.5 / **94.0** | 96.3 / 91.7 / **98.5** | 89.9 / 89.9 / **94.6** | 96.2 / 83.1 / **98.0** | 96.4 / 94.7 / **98.7** |
| | LSUN(R) | 84.5 / 89.0 / **95.7** | 97.1 / 95.1 / **99.0** | 91.4 / 92.1 / **95.4** | 97.2 / 91.0 / **98.9** | 96.8 / 96.7 / **99.0** |
| CIFAR-100 | SVHN | 45.0 / 45.8 / **64.9** | 85.9 / 83.5 / **93.4** | 78.6 / 77.4 / **86.6** | 71.7 / 62.4 / **88.3** | 92.7 / 92.1 / **96.1** |
| | TinyIm(R) | 82.0 / 83.4 / **85.3** | 95.7 / 94.3 / **97.1** | 89.5 / 89.7 / **91.2** | 95.3 / 90.3 / **97.3** | 95.9 / 95.4 / **97.0** |
| | LSUN(R) | 84.5 / **87.7** / 87.2 | 96.6 / 96.6 / **97.5** | 91.0 / 92.0 / **92.4** | 96.8 / 95.7 / **97.8** | 96.1 / 96.6 / **96.9** |
| SVHN | CIFAR-10 | 76.3 / **96.9** / 95.9 | 96.4 / **98.7** / 98.6 | 91.7 / **96.0** / 95.5 | 98.7 / **99.6** / 99.5 | 88.3 / **95.0** / 94.7 |
| | TinyIm(R) | 71.3 / **100.0** / **100.0** | 96.0 / **99.9** / **99.9** | 91.8 / **98.9** / **98.9** | 98.6 / **100.0** / **100.0** | 86.9 / **99.6** / **99.6** |
| | LSUN(R) | 61.4 / **100.0** / **100.0** | 95.1 / **100.0** / 99.9 | 91.0 / **99.3** / 99.2 | 98.3 / **100.0** / **100.0** | 82.3 / **99.7** / 99.6 |

hidden layer. Therefore, $\boldsymbol{f}_l(\boldsymbol{x})$ is a 3D tensor of the size $M_l \times \mathcal{H}_l \times \mathcal{W}_l$ and $\boldsymbol{f}_l^i(\boldsymbol{x})$ is a 2D feature map of the size $\mathcal{H}_l \times \mathcal{W}_l$ for any $i$ and $l$, where $\mathcal{H}_l$ and $\mathcal{W}_l$ are the height and width of each feature map in $l$-th layer, respectively. For a notational convenience, we define $\boldsymbol{m}_l$ as the global average pooling of the $l$-th layer, i.e.,

$$\boldsymbol{m}_l(\boldsymbol{x}) := \|_{i=1}^{M_l} \mathsf{mean}(\boldsymbol{f}_l^i(\boldsymbol{x})),$$

where mean is the average function and $\|$ is the concatenation operation. MAHALANOBIS-vanilla utilizes only the last hidden layer (i.e., the $L$-th layer), so it can be obtained by calculating class means and tied-covariance:

$$\hat{\boldsymbol{\mu}}_k := \frac{1}{N_k} \sum_{\substack{n=1 \\ y_n=k}}^{N} \boldsymbol{m}_L(\boldsymbol{x}_n),$$

$$\hat{\boldsymbol{\Sigma}} := \frac{1}{N} \sum_{k=1}^{K} \sum_{\substack{n=1 \\ y_n=k}}^{N} (\boldsymbol{m}_L(\boldsymbol{x}_n) - \hat{\boldsymbol{\mu}}_k)(\boldsymbol{m}_L(\boldsymbol{x}_n) - \hat{\boldsymbol{\mu}}_k)^T,$$

where $K$ is the number of classes and $N_k$ is the number of the data of class $k$. Then, the out-of-distribution score function of MAHALANOBIS-vanilla detector $\kappa_{\mathrm{mv}}$ (also known as Mahalanobis distance) is defined by

$$\kappa_{\mathrm{mv}}(\boldsymbol{x}^*) := \min_{k=1,\ldots,K} (\boldsymbol{m}_L(\boldsymbol{x}^*) - \hat{\boldsymbol{\mu}}_k)^T \hat{\boldsymbol{\Sigma}}^{-1} (\boldsymbol{m}_L(\boldsymbol{x}^*) - \hat{\boldsymbol{\mu}}_k).$$

In this equation, MAHALANOBIS-vanilla computes the centered data for each class, then performs linear projection for the centered data by using the inverse of $\hat{\boldsymbol{\Sigma}}$. Practically, the inverse of empirical covariance matrix is approximated by (Moore-Penrose) pseudo-inverse, which performs the singular value decomposition (SVD) (Barata & Hussein, 2012). For this reason, computing the Mahalanobis distance becomes equivalent to computing the norm of the sample (i.e., the Euclidean distance from the origin) after the principal component analysis (PCA) assuming that the origin is the class mean, which removes the feature correlations.

### 2.2.2 DEGENERACY OF MAHALANOBIS-ASSEMBLE

From the observation that MAHALANOBIS-vanilla has the power of removing the correlation over feature maps, one may extend this concept to all the other hidden layers. To define the MAHA-LANOBIS-assemble[1] detector, we concatenate the average of feature maps (i.e., global average pooling) of all layers and similarly define the class means and tied-covariance:

$$\boldsymbol{m}(\boldsymbol{x}) := \|_{l=1}^{L} \boldsymbol{m}_l(\boldsymbol{x}),$$

$$\hat{\boldsymbol{\mu}}_k := \frac{1}{N_k} \sum_{\substack{n=1 \\ y_n=k}}^{N} \boldsymbol{m}(\boldsymbol{x}_n), \hat{\boldsymbol{\Sigma}} := \frac{1}{N} \sum_{k=1}^{K} \sum_{\substack{n=1 \\ y_n=k}}^{N} (\boldsymbol{m}(\boldsymbol{x}_n) - \hat{\boldsymbol{\mu}}_k)(\boldsymbol{m}(\boldsymbol{x}_n) - \hat{\boldsymbol{\mu}}_k)^T,$$

$$\kappa_{\mathrm{ma}}(\boldsymbol{x}^*) := \min_{k=1,\ldots,K} (\boldsymbol{m}(\boldsymbol{x}^*) - \hat{\boldsymbol{\mu}}_k)^T \hat{\boldsymbol{\Sigma}}^{-1} (\boldsymbol{m}(\boldsymbol{x}^*) - \hat{\boldsymbol{\mu}}_k)$$

---

[1]We use the term "assemble" to distinguish it from the "feature ensemble", which is the method using logistic regression in the original paper.

From the experiments, we find an interesting observation that the simply extended detector $\kappa_{\mathrm{ma}}$ significantly improve the performance in many cases; however, in some cases, its performance, especially AUPR(In), is poorer than $\kappa_{\mathrm{mv}}$ (see the first two columns of Table 1 for more details).

## 2.3 NORMALIZED COMPRESSION DISTANCE

To address the degeneracy problem in MAHALANOBIS-assemble, we focus on hindering the loss of spatial information of the feature maps caused by the average pooling. To this end, we introduce a compression-based method. Before defining our method precisely, we need to state some notions briefly because its fundamental properties are originated from the concept of information theory.

### 2.3.1 KOLMOGOROV COMPLEXITY

The Kolmogorov complexity (or algorithmic complexity) is the complexity of an arbitrary object which can be represented in string format. Formally, given an universal Turing machine $U$, the Kolmogorov complexity $\mathsf{K}$ of string $x$ is defined as the minimum length of the programs that are decoded into $x$ by $U$. That is,

$$\mathsf{K}(x) := \min\left(\{\mathsf{length}(p)|U(p) = x \text{ where } p \text{ is a valid program for } U\}\right),$$

where $\mathsf{length}(p)$ is the length of the program $p$ and $U(p)$ is the result of decoded string with the program $p$ through $U$.

The conditional Kolmogorov complexity is the minimum program length for a given universal Turing machine which already has the string:

$$\mathsf{K}(x|y) := \min\left(\{\mathsf{length}(p)|U(p|y) = x\}\right).$$

If $x$ is highly correlated with $y$, its conditional complexity would be low.

### 2.3.2 NORMALIZED INFORMATION DISTANCE

The normalized information distance (NID) is the distance between two strings $x$ and $y$ based on the conditional Kolmogorov complexity. It does not need any prior knowledge so that it can be applied to various tasks universally. The distance is described as

$$\mathsf{NID}(x, y) = \frac{\max(\mathsf{K}(x|y), \mathsf{K}(y|x))}{\max(\mathsf{K}(x), \mathsf{K}(y))}.$$

It turns out that NID is a normalized metric (Li et al., 2004), which has following distance properties for arbitrary $x$, $y$, and $z$:

- $\mathsf{NID}(x, y) \in [0, 1]$                  (normality)
- $\mathsf{NID}(x, y) = 0$ if and only if $x = y$      (identity)
- $\mathsf{NID}(x, y) = \mathsf{NID}(y, x)$            (symmetry)
- $\mathsf{NID}(x, z) \leq \mathsf{NID}(x, y) + \mathsf{NID}(y, z)$    (triangle inequality)

Unfortunately, the Kolmogorov complexity is not computable, so NID also cannot be computed. Instead, the normalized compression distance (NCD) is used as the approximation of Kolmogorov complexity in many tasks. Given an arbitrary compressor $C$, NCD is defined by

$$\mathsf{NCD}(x, y) = \frac{\min\left(C(xy), C(yx)\right) - \min\left(C(x), C(y)\right)}{\max(C(x), C(y))},$$

where $\mathsf{C}(x)$ is the length of the compression result of $x$, and $xy$ is the concatenation of $x$ and $y$. Any off-the-shelf compressors can be used as the compressor, including 7-zip, gzip, and bzip. The better compressor we use, the closer the NCD value approaches to the NID value (Cilibrasi & Vitányi, 2005).

## 3 MALCOM: OUT-OF-DISTRIBUTION DETECTOR USING NCD

To resolve the degeneracy of MAHALANOBIS-assemble discussed in Section 2.2.2, we propose the MAhaLanobis distance with COMpressive-complexity pooling (MALCOM), which applies NCD to measure the distance between the feature maps of convolutional neural networks.

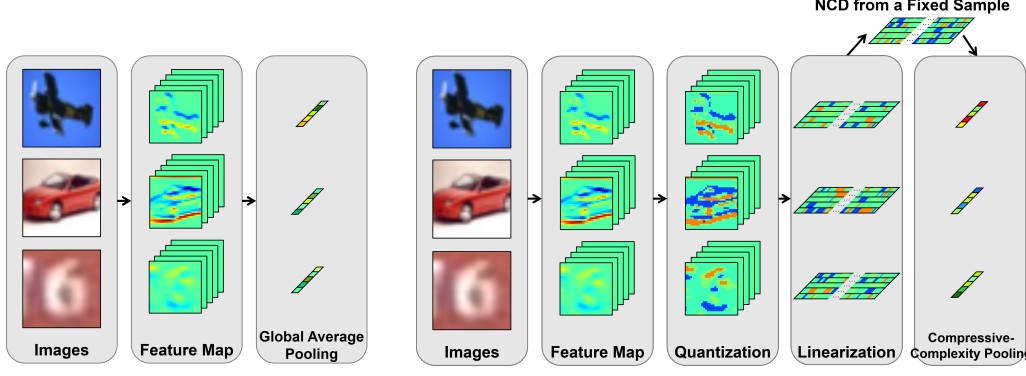

(a) Global average pooling           (b) Compressive-complexity pooling

Figure 1: The proposed compressive-complexity pooling captures the spatial information of feature maps into the final vector by using NCD, whereas global average pooling cannot make use of it.

### 3.1 TNCD

NCD can be generalized well to various objects because of its universality. Most previous work use the raw images (Lan & Harvey) when they calculate NCD, but it is inappropriate to use raw images for many cases. For example, consider the following strings.

$$\text{string } \boldsymbol{a} : \text{BABBABBAABABABBBABAABBABB}$$
$$\text{string } \boldsymbol{b} : \text{CCBACABCCCBAABBABABABAABCBA}$$
$$\text{string } \boldsymbol{c} : \text{ABACABACAABCBAACBBBCABACA}$$

In case that we only want to recognize the pattern of the symbol C, using NCD on this string is not useful enough. It needs to look at the features that the classifier consider important. To focus on such important features, we introduce a translation $T$. In the case above, for example, $T$ translates the symbol $C$ into $1$ and all other alphabet into $0$.

$$T(\boldsymbol{a}) : 0000000000000000000000000$$
$$T(\boldsymbol{b}) : 1100100111000000000000100$$
$$T(\boldsymbol{c}) : 0001000100010001000100010$$

Before the translation, string $\boldsymbol{a}$ looks much simpler than string $\boldsymbol{b}$ or string $\boldsymbol{c}$. Through the transformation, we can concentrate only on $C$ in the string $\boldsymbol{b}$ and string $\boldsymbol{c}$. Formally, for the two symbol sets $\Sigma$ and $\Omega$, we denote the translation by $T : \Sigma^* \rightarrow \Omega^*$. Given a transformation $T$, we define the translated and normalized compression distance (tNCD) by

$$\text{tNCD}(\boldsymbol{x}, \boldsymbol{y}; T) = \frac{\min\left[C(T(\boldsymbol{x})T(\boldsymbol{y})), C(T(\boldsymbol{y})T(\boldsymbol{x}))\right] - \min\left[C(T(\boldsymbol{x})), C(T(\boldsymbol{y}))\right]}{\max\left[C\left(T(\boldsymbol{x})\right), C\left(T(\boldsymbol{y})\right)\right]}.$$

Note that tNCD between $\boldsymbol{x}$ and $\boldsymbol{y}$ depends on what the translation $T$ focuses on. If we fix $\boldsymbol{y}$ and there exist a lot of translations with different perspectives, $\boldsymbol{x}$ can be described by various aspects $\boldsymbol{y}$.

### 3.2 COMPRESSIVE-COMPLEXITY POOLING

We propose the *compressive-complexity pooling* which utilize the spatial information of a feature map. For each feature map $\boldsymbol{f}_l^i$, we consider $\boldsymbol{h}_l^i := \boldsymbol{g}_l^i \circ \boldsymbol{f}_l^i$ as a translation where $\boldsymbol{g}_l^i$ is a quantization-and-linearization, which discretizes the continuous values and then makes the quantized feature map a string. We first select a sample $\boldsymbol{s}$ from $\mathcal{D}_{\text{train}}$. Then, for the fixed sample $\boldsymbol{s}$ and the test sample $\boldsymbol{x}$, we can calculate the tNCD value $d_l^i(\boldsymbol{x}; \boldsymbol{s})$ for each translation $h_l^i$,

$$d_l^i(\boldsymbol{x}; \boldsymbol{s}) := \text{tNCD}(\boldsymbol{s}, \boldsymbol{x}; h_l^i) = \frac{\min\left[C(h_l^i(\boldsymbol{x})h_l^i(\boldsymbol{y})), C(h_l^i(\boldsymbol{y})h_l^i(\boldsymbol{x}))\right] - \min\left[C(h_l^i(\boldsymbol{x})), C(h_l^i(\boldsymbol{y}))\right]}{\max\left[C\left(h_l^i(\boldsymbol{x})\right), C\left(h_l^i(\boldsymbol{y})\right)\right]}.$$

Intuitively, for an image $\boldsymbol{x}$ drawn from a specific class, the distance $d_l^i$ becomes small if some aspects of $\boldsymbol{x}$ corresponding to the translation $h_l^i$ are similar to $\boldsymbol{s}$, and it would be large otherwise.

The compressive-complexity pooling is defined by a fixed sample $\boldsymbol{s}$ with aggregating every feature maps,

$$\boldsymbol{\rho}_l(\boldsymbol{x}; \boldsymbol{s}) = \|_{i=1}^{M_l} d_l^i(\boldsymbol{x}, \boldsymbol{s}).$$

All the layers can be concatenated for $\boldsymbol{x}$ with respect to $\boldsymbol{s}$,

$$\boldsymbol{\rho}(\boldsymbol{x}; \boldsymbol{s}) := \|_{l=1}^{L} \boldsymbol{\rho}_l(\boldsymbol{x}; \boldsymbol{s}).$$

This can be interpreted as the distance of $\boldsymbol{x}$ from $\boldsymbol{s}$, considering the patterns of spatial information for each feature map. Figure 1 illustrates the overall process of the compressive-complexity pooling and how it differs from the global average pooling.

### 3.3 SCORE FUNCTION FOR OUT-OF-DISTRIBUTION

To put it all together, we can obtain the mean vectors $\tilde{\boldsymbol{m}}$ that encodes spatial information by concatenating the mean pooling vectors and the compressive-complexity pooling vectors. We apply the Mahalanobis distance to remove correlation between feature maps.

$$\tilde{\boldsymbol{m}}(\boldsymbol{x}) := \boldsymbol{m}(\boldsymbol{x}) \parallel \boldsymbol{\rho}(\boldsymbol{x}; \boldsymbol{s})$$

$$\hat{\boldsymbol{\mu}}_k := \frac{1}{N_k} \sum_{\substack{n=1 \\ y_n=k}}^{N} \tilde{\boldsymbol{m}}(\boldsymbol{x}_n), \hat{\boldsymbol{\Sigma}} := \frac{1}{N} \sum_{k=1}^{K} \sum_{\substack{n=1 \\ y_n=k}}^{N} (\tilde{\boldsymbol{m}}(\boldsymbol{x}_n) - \hat{\boldsymbol{\mu}}_k)(\tilde{\boldsymbol{m}}(\boldsymbol{x}_n) - \hat{\boldsymbol{\mu}}_k)^T$$

Based on the mean vectors, we define the final out-of-distribution score function $\kappa$ which is capable of accurately measuring the out-of-distribution score:

$$\kappa_{\text{MALCOM}}(\boldsymbol{x}^*) := \min_{k=1,\ldots,K} (\tilde{\boldsymbol{m}}(\boldsymbol{x}^*) - \hat{\boldsymbol{\mu}}_k)^T \hat{\boldsymbol{\Sigma}}^{-1} (\tilde{\boldsymbol{m}}(\boldsymbol{x}^*) - \hat{\boldsymbol{\mu}}_k).$$

## 4 EXPERIMENTS

### 4.1 EXPERIMENTAL SETUP

For experiments, we follow the experimental setup mainly used by previous work (Liang et al., 2018; Lee et al., 2018b).

**Models.** For fair comparisons, we use the pretrained DenseNet (Huang et al., 2017) and ResNet (He et al., 2016) provided by the previous work (Lee et al., 2018b)[2]. According to the authors, they use DenseNet with 100 layers, growth rate k=12 and dropout rate 0, while ResNet with 34 layers and dropout rate 0. Both the models are trained using stochastic gradient descent (SGD) with Nesterov momentum for classifying CIFAR-10, CIFAR-100 (Krizhevsky et al., 2009) and SVHN (Netzer et al., 2011). For more details, we refer to the original paper (Lee et al., 2018b).

**Dataset.** We use out-of-distribution image datasets: TinyImageNet (Deng et al., 2009), LSUN (Yu et al., 2015), and iSUN (Xu et al., 2015). The in-distribution datasets have 10 classes except for CIFAR-100 which has 100 classes. CIFAR-10 (or CIFAR-100) has 50,000 training and 10,000 test images, while SVHN has 73,257 training and 26,032 test images. For out-of-distribution datasets, TinyImageNet contains 10,000 test images of 200 classes, LSUN contains 10,000 of 10 classes, and iSUN contains 8925 images. Liang et al. (2018)[3] preprocessed TinyImageNet and LSUN by either randomly cropping of size $32 \times 32$ or downsampling to size $32 \times 32$.

**Competing Methods.** We compare the performance of our method with that of the existing methods satisfying the two constraints (Section 2.1), which are the independence with out-of-distribution samples and exploitation of the model without retraining. As far as we know, there is only one method perfectly satisfying these requirements: BASELINE (Hendrycks & Gimpel, 2017) using the

---

[2]https://github.com/pokaxpoka/deep_Mahalanobis_detector
[3]https://github.com/facebookresearch/odin

Table 2: Ablation results on analyzing the effect of each component: the concatenation of feature maps from all layers (denoted by layer aggregation), global average pooling, and compressive-complexity pooling. AUROC is used as an evaluation metric. The best results are marked in boldface.

| Method | Layer aggregation | Global average pooling | Compressive complexity pooling | CIFAR-10 | | CIFAR-100 | |
|---|---|---|---|---|---|---|---|
| | | | | TinyIm | LSUN | TinyIm | LSUN |
| BASELINE | - | - | - | 94.14 | 95.47 | 71.62 | 70.84 |
| MAHALANOBIS-vanilla | - | ✓ | - | 96.27 | 97.06 | 95.69 | 96.60 |
| MAHALANOBIS-assemble | ✓ | ✓ | - | 91.67 | 95.09 | 94.29 | 96.59 |
| MALCOM (ours) | - | - | ✓ | 93.26 | 94.42 | 92.29 | 92.28 |
| | - | ✓ | ✓ | 95.47 | 96.14 | 95.92 | 96.78 |
| | ✓ | - | ✓ | 95.77 | 95.33 | 93.31 | 92.27 |
| | ✓ | ✓ | ✓ | **98.52** | **98.96** | **97.11** | **97.47** |

softmax output to distinguish out-of-distribution data from in-distribution data. In addition, we can use the simplified version of (Lee et al., 2018b), i.e., MAHALANOBIS-vanilla (Section 2.2.1), although the original method uses the input preprocessing and ensembles the layers by introducing the hyperparameters. The implementation details about the compression of the feature map are provided in Appendix A.

**Evaluation Metrics.** We adopt the five performance metrics the same with the previous work (Lee et al., 2018b): true negative rate (TNR) at 95% true positive rate (TPR), area under the receiver operating characteristic curve (AUROC), area under the precision-recall curve (AUPR), and detection accuracy. For the details, we refer to (Lee et al., 2018b).

## 4.2 RESULTS

**Ablation study.** We ablate the components of our method to show the interaction between the global average pooling and the compressive-complexity pooling. To this end, we use the DenseNet pretrained on CIFAR-10 and CIFAR-100. The out-of-distribution datasets are the resized TinyImageNet and the resized LSUN, and the AUROC is measured as the evaluation metric. Table 2 shows the results of the ablation study. The layer aggregation refers to the method that uses the concatenation of feature maps of all the layers. From the results, we obtain the following observations:

- The performance of our compressive-complexity pooling is improved by concatenating the feature maps of all the layers, whereas the performance of the global average pooling becomes worse when it concatenates the feature maps of all the layers.

- The compressive-complexity pooling is always worse than the global average pooling when we use only the feature maps of last hidden layer.

- Using both the pooling techniques at the same time consistently improves the detection performance. It shows that each pooling provides different information.

**Main results.** The main results are summarized in Table 3. we repeat each experiment five times and report the averaged results. See Appendix D for more detailed results. (C) and (R) denote the dataset "resized" and "cropped", respectively. Our proposed method, MALCOM, achieves significantly better performance than other competing methods in most cases. Nevertheless, for minor cases, we observe that MALCOM with ResNet slightly degrades the performance on the resized TinyImageNet and the resized LSUN. From the extensive experiments, we conclude that our compressive-complexity pooling successfully captures the spatial information from its feature maps, and it is helpful to accurately measure the confidence score for detecting the out-of-distribution data.

## 5 RELATED WORK

As deep learning has made remarkable progress in the past few years, many methods have been proposed to estimate the confidence of the model prediction beyond just enhancing the model accuracy. Although the modern neural networks have achieved higher classification accuracy than in the

Table 3: Performance of different out-of-distribution detection methods using pretrained DenseNet and ResNet. For out-of-distribution datasets, 'C' and 'R' are the abbreviations for 'cropped' and 'resized', respectively. The best result is highlighted by boldface.

| ID | | OOD | TNR at TPR 95% | AUROC | Detection Acc. | AUPR(In) | AUPR(Out) |
|---|---|---|---|---|---|---|---|
| | | | BASELINE / MAHALANOBIS-vanilla / MALCOM (ours) | | | | |
| DenseNet | CIFAR-10 | SVHN | 40.4 / 89.2 / **93.4** | 89.9 / 97.6 / **98.4** | 83.2 / 92.4 / **94.3** | 83.7 / 94.5 / **95.9** | 94.4 / 99.0 / **99.3** |
| | | TinyIm(C) | 65.8 / 74.6 / **100.0** | 95.4 / 95.4 / **99.9** | 90.1 / 88.4 / **99.1** | 96.5 / 95.8 / **99.9** | 93.9 / 95.1 / **99.9** |
| | | TinyIm(R) | 59.4 / 82.3 / **94.0** | 94.1 / 96.3 / **98.5** | 88.5 / 90.0 / **94.6** | 95.3 / 96.2 / **98.0** | 92.4 / 96.4 / **98.7** |
| | | LSUN(C) | 61.4 / 52.9 / **99.9** | 94.9 / 90.8 / **99.8** | 89.8 / 82.8 / **98.6** | 96.2 / 91.9 / **99.8** | 93.2 / 89.6 / **99.8** |
| | | LSUN(R) | 65.8 / 74.6 / **100.0** | 95.4 / 95.4 / **99.9** | 90.1 / 88.4 / **99.1** | 96.5 / 95.8 / **99.9** | 93.9 / 95.1 / **99.9** |
| | CIFAR-100 | SVHN | 26.2 / 45.0 / **64.9** | 82.6 / 85.9 / **93.4** | 75.5 / 78.6 / **86.6** | 75.1 / 71.7 / **88.3** | 90.4 / 92.7 / **96.1** |
| | | TinyIm(C) | 32.7 / 52.9 / **98.1** | 81.1 / 86.8 / **98.8** | 72.9 / 79.1 / **96.6** | 82.9 / 85.9 / **99.2** | 80.0 / 87.3 / **98.2** |
| | | TinyIm(R) | 17.3 / 82.0 / **85.3** | 71.6 / 95.7 / **97.1** | 65.7 / 89.5 / **91.2** | 74.1 / 95.3 / **97.3** | 69.0 / 95.9 / **97.0** |
| | | LSUN(C) | 40.1 / 12.4 / **95.3** | 85.4 / 60.6 / **98.4** | 76.7 / 56.9 / **95.7** | 87.0 / 61.7 / **98.8** | 84.4 / 60.5 / **97.2** |
| | | LSUN(R) | 16.4 / 84.5 / **87.2** | 70.8 / 96.6 / **97.5** | 65.0 / 91.0 / **92.4** | 74.2 / 96.8 / **97.8** | 67.9 / 96.1 / **96.9** |
| | SVHN | CIFAR-10 | 69.1 / 76.3 / **95.9** | 91.8 / 96.4 / **98.6** | 86.5 / 91.7 / **95.5** | 95.4 / 98.7 / **99.5** | 83.5 / 88.3 / **94.7** |
| | | TinyIm(C) | 79.0 / 64.8 / **100.0** | 94.8 / 95.0 / **100.0** | 89.9 / 90.5 / **100.0** | 97.2 / 98.3 / **100.0** | 89.1 / 83.1 / **100.0** |
| | | TinyIm(R) | 79.7 / 71.3 / **100.0** | 94.8 / 96.0 / **99.9** | 90.2 / 91.8 / **98.9** | 97.0 / 98.6 / **100.0** | 88.9 / 86.9 / **99.6** |
| | | LSUN(C) | 76.7 / 70.6 / **100.0** | 93.8 / 95.3 / **100.0** | 88.8 / 89.7 / **100.0** | 96.7 / 98.2 / **100.0** | 87.4 / 86.8 / **100.0** |
| | | LSUN(R) | 77.1 / 61.4 / **100.0** | 94.1 / 95.1 / **99.9** | 89.2 / 91.0 / **99.2** | 96.7 / 98.3 / **100.0** | 88.0 / 82.3 / **99.6** |
| ResNet | CIFAR-10 | SVHN | 32.2 / 54.5 / **79.2** | 89.9 / 93.9 / **96.6** | 85.1 / 89.1 / **91.1** | 85.9 / 91.6 / **94.4** | 93.6 / 96.0 / **98.3** |
| | | TinyIm(C) | 49.1 / 66.1 / **99.0** | 92.6 / 95.4 / **99.6** | 86.9 / 90.4 / **97.3** | 94.1 / 96.6 / **99.7** | 90.1 / 93.3 / **99.6** |
| | | TinyIm(R) | 44.1 / 69.3 / **89.7** | 91.0 / 95.0 / **98.3** | 85.0 / 88.6 / **93.1** | 92.5 / 95.8 / **98.4** | 88.3 / 94.0 / **98.2** |
| | | LSUN(C) | 52.8 / 50.1 / **98.1** | 93.3 / 94.3 / **99.5** | 88.1 / 90.3 / **96.7** | 94.8 / 96.0 / **99.6** | 90.9 / 90.8 / **99.4** |
| | | LSUN(R) | 45.1 / 78.3 / **91.6** | 91.1 / 96.6 / **98.5** | 85.3 / 90.7 / **93.8** | 92.5 / 97.1 / **98.7** | 88.6 / 95.8 / **98.4** |
| | CIFAR-100 | SVHN | 19.9 / 43.2 / **61.4** | 79.3 / 89.4 / **92.6** | 73.2 / 82.0 / **84.4** | 65.9 / 83.6 / **87.2** | 88.3 / 94.3 / **96.4** |
| | | TinyIm(C) | 16.9 / 34.7 / **88.8** | 75.8 / 85.5 / **97.9** | 70.1 / 78.2 / **92.1** | 76.5 / 87.1 / **97.9** | 71.2 / 82.6 / **97.9** |
| | | TinyIm(R) | 20.2 / 20.3 / **64.2** | 77.1 / 78.9 / **92.5** | 70.8 / 72.0 / **83.9** | 79.7 / 81.8 / **93.0** | 73.2 / 74.4 / **92.5** |
| | | LSUN(C) | 12.7 / 42.5 / **88.5** | 70.5 / 87.4 / **97.8** | 66.6 / 80.0 / **92.0** | 68.1 / 88.4 / **97.8** | 66.5 / 85.3 / **97.8** |
| | | LSUN(R) | 18.4 / 19.6 / **60.6** | 75.6 / 79.0 / **91.6** | 69.8 / 72.3 / **82.7** | 77.4 / 82.0 / **92.4** | 71.7 / 74.1 / **91.3** |
| | SVHN | CIFAR-10 | 78.3 / 85.0 / **90.4** | 93.0 / 97.0 / **98.0** | 90.1 / 93.1 / **94.5** | 94.8 / 99.0 / **99.3** | 86.4 / 89.3 / **93.4** |
| | | TinyIm(C) | 82.5 / 83.3 / **100.0** | 94.0 / 96.9 / **100.0** | 91.4 / 93.9 / **99.1** | 95.3 / 99.0 / **100.0** | 88.4 / 86.8 / **99.9** |
| | | TinyIm(R) | 79.1 / 84.5 / **97.7** | 93.5 / 97.0 / **99.2** | 90.4 / 93.1 / **96.4** | 95.4 / 98.9 / **99.7** | 86.9 / 89.1 / **97.8** |
| | | LSUN(C) | 79.6 / 84.1 / **100.0** | 93.1 / 97.0 / **99.9** | 90.5 / 93.2 / **98.5** | 94.5 / 99.0 / **100.0** | 87.3 / 87.8 / **99.7** |
| | | LSUN(R) | 74.5 / 78.3 / **97.2** | 91.5 / 96.2 / **99.0** | 88.9 / 91.9 / **96.2** | 93.8 / 98.7 / **99.7** | 84.6 / 85.9 / **97.1** |

past, the confidence calibration in neural networks is poorer than a decade ago, i.e., the discrepancy between the prediction score of the neural network and the ground-truth class probability gets worse compared with that of the past (Guo et al., 2017). To make matters worse, most neural networks, especially for non-Bayesian, have no measure of uncertainty, which hinders reliable decision-making just by the prediction score in many practical areas (e.g. medical diagnosis, self-driving car). Thus, there has been much work to attain the measurement for uncertainty or confidence calibration other than classification accuracy (Guo et al., 2017; Lakshminarayanan et al., 2017; Jiang et al., 2018)

Even though Bayesian deep learning provides uncertainty properly, several methods were recently proposed to estimate uncertainty of non-Bayesian deep neural networks. Especially, it turns out that some regularization techniques of neural networks (e.g. dropout, batch normalization) are interpreted as approximate Bayesian inference so that uncertainty can be captured in conventional deep neural network (Gal, 2016; Teye et al., 2018). These work are significant in that deep neural networks are not required to re-model or retrain once they have been trained by using dropout or batch normalization even if they do not consider the uncertainty at train time. MALCOM is inspired mainly by these work because it can be applied more practically to neural networks which already have been trained and deployed.

Most previous work of out-of-distribution detection can be divided into two main approaches: introducing parameter and defining new objective function. An prominent example of the former is Liang et al. (2018) (ODIN) which suggests two hyperparameters for softmax output to a boost in performance: temperature scaling and input preprocessing. Since ODIN, most studies using softmax output use these two parameters. Vyas et al. (2018) tried to solve the problem with ensemble of classifiers by dividing the training data and adding loss by defining entropy for in-distribution and out-of-distribution as a self-supervised learning method, and used two parameters to improve performance. Our closely related work, Lee et al. (2018b) focuses on the internal features of the

CNNs, while it uses some hyperparameter logistic regression for aggregating all feature maps and input preprocessing.

## 6 CONCLUSION

This paper proposes MALCOM, which accurately measures the confidence score of a test sample for out-of-distribution detection, without using any hyperparameters and without retraining the model. MALCOM addresses the limitation of the existing Mahalanobis distance-based method that cannot exploit the spatial information of image feature maps. We introduce the compressive-complexity pooling based on NCD to consider the spatial information of the feature map when measuring the distance from class means. Our extensive evaluation on image benchmark datasets shows that the compressive-complexity pooling leverages the spatial information of the feature maps, and it improve the performance compared to the global average pooling. In addition, by using the both the pooling techniques simultaneously, MALCOM achieves the best performance for out-of-distribution detection.

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

# A    IMPLEMENTATION DETAILS

**Lempel-Ziv-Welch complexity.**    Although there are many efficient off-the-shelf compressing algorithms (e.g. 7-zip, GZip, and BZip), we do not consider them because strings do not need to be actually compressed. We only need to obtain the lengths of compressing results to calculate NCD. Because of its computational efficiency, we adopt the Lemple-Ziv-Welch (LZW) coding scheme (Welch, 1984), which enables to compute the length of a string in $O(n)$ and to accelerate the computation by using GPU.

**Hilbert space-filling curve.**    There are several choices to linearize the 2D feature maps, including simple row-wise (or column-wise) linearization. Among them, linearizing the images by Hilbert curve turns out to be effective when using NCD for images (Liang et al., 2008; Mortensen et al., 2009; Coltuc et al., 2018). For this reason, we adopt Hilbert curve to linearize the 2D feature maps.

**Lloyd-Max quantizer.**    We quantize the feature maps by using the Lloyd-Max quantizer (Lloyd, 1982; Max, 1960), which effectively minimizes the information loss. We fix the quantization level to 4, which is known to perform well for images when it used with our compression scheme (i.e., LZW) (Pinho & Ferreira, 2011; Coltuc et al., 2018).

# B    VALIDATION ON OUT-OF-DISTRIBUTION SAMPLES

In this section, we compare our proposed method with the two additional baselines: ODIN (Liang et al., 2018) and MAHALANOBIS (Lee et al., 2018b). Both of them utilize out-of-distribution samples to find the optimal hyperparameters values. In this sense, they do not satisfy our first constraint describing that any test distributions should not be assumed, as discussed in Section 2.1. Nevertheless, we loosen this constraint and report the results to show relative the performance compared to the state-of-the-art methods.

**ODIN.**    ODIN increases the maximum value of the softmax score using temperature scaling and input preprocessing, and uses this softmax score as the score function. Let $\boldsymbol{f}_{L+1} \in \mathbb{R}^K$ be the outputs of the last fully-connected layer of a trained model $\boldsymbol{f}$, i.e., $\boldsymbol{f}_{L+1} = \left[ f_{L+1}^1, \cdots, f_{L+1}^K \right]$ is the layer right before applying softmax. Then, with the two hyperparameters $T_{\mathrm{o}}$ and $\epsilon_{\mathrm{o}}$, respectively for the temperature scaling and the input preprocessing, the out-of-distribution score function is defined as follows:

$$\sigma_k(\boldsymbol{z}; T_{\mathrm{o}}) := \frac{e^{z_k/T_{\mathrm{o}}}}{\sum_{j=1}^K e^{z_j/T_{\mathrm{o}}}} \qquad \forall \boldsymbol{z} = [z_1, \cdots, z_K] \in \mathbb{R}^K,$$

$$S(\boldsymbol{x}; T_{\mathrm{o}}) := \max_{k=1,\ldots,K} \sigma_k \left( \boldsymbol{f}_{L+1}\left( \boldsymbol{z} \right); T_{\mathrm{o}} \right),$$

$$\kappa_{\mathrm{ODIN}}(\boldsymbol{x}^*; \epsilon_o, T_{\mathrm{o}}) := -S \left( \boldsymbol{x}^* + \epsilon_{\mathrm{o}} \mathsf{sign} \left( \nabla \log S(\boldsymbol{x}^*; T_{\mathrm{o}}) \right); T_{\mathrm{o}} \right),$$

where sign is the sign function.

**Mahalanobis.**    MAHALANOBIS is similar to our proposed method in that it calculates the Mahalanobis distance by using the global average pooling on feature maps from a target layer. The main difference of MAHALANOBIS from MAHALANOBIS-vanilla is that it uses the weighted sum of multiple Mahalanobis scores, which are obtained from more than one layers, and the weight for the $l$-th score is introduced as the hyperparameter $\alpha_{\mathrm{m}}^l$. In addition, MAHALANOBIS also adopts the input preprocessing technique with the hyperparameter $\epsilon_{\mathrm{m}}$, which plays a similar role to ODIN. The

Table 4: Performance of different out-of-distribution detection methods that utilize out-of-distribution samples for the validation. For out-of-distribution datasets, 'C' and 'R' are the abbreviations for 'cropped' and 'resized', respectively. The best result is highlighted in boldface.

| ID | | OOD | TNR at TPR 95% | AUROC | Detection Acc. |
|---|---|---|---|---|---|
| | | | ODIN / Mahalanobis / MALCOM (ours) / MALCOM++(ours) | | |
| DenseNet | CIFAR-10 | SVHN | 86.16 / 92.35 / 93.67 / **95.07** | 95.51 / 98.06 / 98.52 / **98.84** | 91.43 / 93.72 / 94.45 / **95.05** |
| | | TinyIm(C) | 95.70 / 99.36 / **99.99** / 99.53 | 99.11 / 99.82 / **99.92** / 99.89 | 95.36 / 98.93 / **99.17** / 99.04 |
| | | TinyIm(R) | 9292.41 / 95.04 / 94.38 / **95.41** | 98.51 / 98.89 / 98.67 / **99.03** | 93.91 / 95.05 / 94.74 / **95.33** |
| | | LSUN(C) | 91.08 / 98.87 / **99.79** / 99.11 | 98.21 / 99.63 / 99.78 / **99.79** | 94.02 / 98.62 / 98.65 / **98.66** |
| | | LSUN(R) | 96.22 / **97.04** / 95.79 / 96.89 | 99.23 / 99.21 / 98.98 / **99.25** | 95.66 / **96.14** / 95.44 / 96.11 |
| | | iSUN | 93.87 / 95.21 / 94.07 / **95.64** | 98.84 / 98.92 / 98.75 / **99.09** | 94.56 / 95.14 / 94.61 / **95.41** |
| | CIFAR-100 | SVHN | 70.65 / **87.23** / 64.80 / 85.79 | 93.82 / **97.08** / 93.47 / 96.27 | 86.58 / **91.23** / 86.76 / 90.68 |
| | | TinyIm(C) | 72.51 / **99.38** / 98.04 / 98.65 | 94.35 / **99.78** / 98.82 / 99.54 | 86.93 / **98.96** / 96.60 / 98.28 |
| | | TinyIm(R) | 42.64 / **88.38** / 85.64 / 88.36 | 85.22 / 97.43 / 97.17 / **97.51** | 77.04 / **92.18** / 91.43 / 92.17 |
| | | LSUN(C) | 81.18 / **99.48** / 95.51 / 98.67 | 96.49 / **99.76** / 98.31 / 99.59 | 90.17 / **98.86** / 95.72 / 98.14 |
| | | LSUN(R) | 41.21 / 92.13 / 88.54 / **92.44** | 85.45 / 97.83 / 97.67 / **97.95** | 77.06 / 93.65 / 92.89 / **93.76** |
| | | iSUN | 37.41 / 88.43 / 85.28 / **89.26** | 84.06 / 97.32 / 97.26 / **97.60** | 76.04 / 92.22 / 91.61 / **92.44** |
| | SVHN | CIFAR-10 | 71.73 / **96.53** / 95.40 / 94.61 | 91.37 / **99.02** / 98.56 / 98.57 | 85.80 / **96.32** / 95.44 / 95.02 |
| | | TinyIm(C) | 81.76 / **100.00 / 100.00 / 100.00** | 94.84 / **100.00 / 100.00** / 99.99 | 89.64 / **99.99 / 99.99** / 99.97 |
| | | TinyIm(R) | 84.06 / 99.57 / **99.96** / 99.51 | 95.11 / **99.88** / 99.87 / 99.86 | 90.35 / **98.93 / 98.93** / 98.84 |
| | | LSUN(C) | 74.41 / **100.00 / 100.00 / 100.00** | 91.89 / **100.00 / 100.00 / 100.00** | 86.38 / **99.98 / 99.98 / 99.98** |
| | | LSUN(R) | 81.12 / 99.65 / **100.00** / 99.61 | 94.54 / **99.90** / 99.89 / 99.88 | 89.16 / **99.29** / 99.26 / 99.16 |
| | | iSUN | 82.15 / 99.64 / **99.99** / 99.88 | 94.68 / **99.89** / 99.88 / 99.88 | 89.62 / **99.31** / 99.17 / 99.18 |
| ResNet | CIFAR-10 | SVHN | 86.55 / **96.19** / 78.61 / 96.15 | 96.65 / 99.14 / 96.56 / **99.19** | 91.08 / **95.76** / 91.13 / 95.65 |
| | | TinyIm(C) | 74.76 / 99.85 / 98.86 / **99.90** | 94.58 / 99.97 / 99.59 / **99.98** | 87.39 / 99.29 / 97.20 / **99.50** |
| | | TinyIm(R) | 72.51 / 97.38 / 89.52 / **97.64** | 94.04 / 99.47 / 98.27 / **99.52** | 86.48 / 96.32 / 93.16 / **96.51** |
| | | LSUN(C) | 72.08 / 99.86 / 97.95 / **99.89** | 93.80 / **99.97** / 99.47 / 99.97 | 86.88 / 99.30 / 96.56 / **99.43** |
| | | LSUN(R) | 73.83 / 98.74 / 91.37 / **98.99** | 94.14 / 99.69 / 98.52 / **99.72** | 86.69 / 97.49 / 93.76 / **97.55** |
| | | iSUN | 73.20 / 97.96 / 91.02 / **98.15** | 94.02 / 99.52 / 98.45 / **99.56** | 86.49 / 96.74 / 93.64 / **96.93** |
| | CIFAR-100 | SVHN | 62.75 / **92.41** / 61.63 / 91.41 | 93.94 / **98.18** / 92.60 / 97.98 | 88.05 / **93.75** / 84.62 / 93.30 |
| | | TinyIm(C) | 30.82 / 99.74 / 88.70 / **99.77** | 79.75 / 99.91 / 97.81 / **99.92** | 73.51 / 99.23 / 92.04 / **99.29** |
| | | TinyIm(R) | 49.19 / 89.44 / 64.61 / **92.31** | 87.62 / 97.93 / 92.60 / **98.38** | 80.11 / 92.74 / 84.04 / **93.75** |
| | | LSUN(C) | 37.47 / 99.64 / 88.21 / **99.76** | 76.17 / 99.89 / 97.67 / **99.91** | 71.03 / 99.05 / 91.78 / **99.31** |
| | | LSUN(R) | 45.59 / 92.80 / 61.51 / **92.89** | 85.64 / 98.27 / 91.77 / **98.32** | 78.26 / 93.93 / 82.94 / **93.97** |
| | | iSUN | 45.26 / 89.77 / 61.63 / **90.94** | 85.45 / 97.85 / 91.54 / **98.04** | 78.41 / 92.87 / 82.62 / **93.16** |
| | SVHN | CIFAR-10 | 79.83 / **97.62** / 90.26 / 97.51 | 92.09 / 99.31 / 98.00 / **99.35** | 89.44 / **96.91** / 94.44 / 96.87 |
| | | TinyIm(C) | 84.69 / 99.97 / **100.00** / 99.97 | 93.75 / 99.98 / 99.96 / **99.99** | 91.23 / 99.94 / 99.13 / **99.96** |
| | | TinyIm(R) | 82.10 / 99.73 / 97.56 / **99.74** | 91.99 / 99.88 / 99.19 / **99.90** | 89.35 / **99.08** / 96.34 / 99.06 |
| | | LSUN(C) | 82.41 / 99.94 / **100.00** / 99.95 | 92.72 / **99.98** / 99.90 / 99.98 | 90.28 / 99.91 / 98.58 / **99.93** |
| | | LSUN(R) | 77.69 / **99.84** / 96.96 / 99.84 | 89.44 / 99.89 / 99.00 / **99.91** | 87.35 / 99.52 / 96.22 / **99.53** |
| | | iSUN | 79.10 / **99.79** / 97.24 / 99.78 | 91.32 / **99.93** / 99.10 / 99.92 | 89.22 / **99.48** / 96.28 / 99.36 |

formulation of Mahalanobis is given by

$$\hat{\boldsymbol{\mu}}_{l,k} := \frac{1}{N_k} \sum_{\substack{n=1 \\ y_n=k}}^{N} \boldsymbol{m}_l(\boldsymbol{x}_n), \quad \hat{\boldsymbol{\Sigma}}_l := \frac{1}{N} \sum_{k=1}^{K} \sum_{\substack{n=1 \\ y_n=k}}^{N} (\boldsymbol{m}_l(\boldsymbol{x}_n) - \hat{\boldsymbol{\mu}}_{l,k})(\boldsymbol{m}_l(\boldsymbol{x}_n) - \hat{\boldsymbol{\mu}}_{l,k})^T,$$

$$M_l(\boldsymbol{x}) := \min_{k=1,\ldots,K} (\boldsymbol{m}_l(\boldsymbol{x}) - \hat{\boldsymbol{\mu}}_{l,k})^T \hat{\boldsymbol{\Sigma}}_l^{-1} (\boldsymbol{m}_l(\boldsymbol{x}) - \hat{\boldsymbol{\mu}}_{l,k}),$$

$$\kappa_{\text{Mahalanobis}}(\boldsymbol{x}^*; \boldsymbol{\alpha}_\text{m}, \epsilon_\text{m}) = \sum_l \alpha_\text{m}^l M_l\left(\boldsymbol{x}^* + \epsilon_\text{m}\text{sign}\left(\nabla M_l(\boldsymbol{x}^*)\right)\right),$$

where $\boldsymbol{m}_l$ is the global average pooling of the $l$-th layer as defined in Section 2.2.1.

**MALCOM++.** We build MALCOM++ by simply extending our proposed method to use out-of-distribution samples for the validation. It uses the weighted sum of the MALCOM scores obtained

from multiple layers, similarly to MAHALANOBIS.

$$\tilde{\boldsymbol{m}}_l(\boldsymbol{x}) := \boldsymbol{m}_l(\boldsymbol{x}) \parallel \boldsymbol{\rho}_l(\boldsymbol{x}; \boldsymbol{s}),$$

$$\tilde{\boldsymbol{\mu}}_{l,k} := \frac{1}{N_k} \sum_{\substack{n=1 \\ y_n=k}}^{N} \tilde{\boldsymbol{m}}_l(\boldsymbol{x}_n), \quad \tilde{\boldsymbol{\Sigma}}_l := \frac{1}{N} \sum_{k=1}^{K} \sum_{\substack{n=1 \\ y_n=k}}^{N} (\tilde{\boldsymbol{m}}_l(\boldsymbol{x}_n) - \tilde{\boldsymbol{\mu}}_{l,k})(\tilde{\boldsymbol{m}}_l(\boldsymbol{x}_n) - \tilde{\boldsymbol{\mu}}_{l,k})^T,$$

$$\tilde{M}_l(\boldsymbol{x}) := \min_{k=1,\ldots,K} (\boldsymbol{m}_l(\boldsymbol{x}) - \hat{\boldsymbol{\mu}}_{l,k})^T \hat{\boldsymbol{\Sigma}}_l^{-1} (\boldsymbol{m}_l(\boldsymbol{x}) - \hat{\boldsymbol{\mu}}_{l,k}),$$

$$\kappa_{\text{MALCOM++}}(\boldsymbol{x}^*; \boldsymbol{\alpha}_c) = \sum_{l=1} \alpha_c^l \tilde{M}_l(\boldsymbol{x}^*),$$

where $\boldsymbol{\rho}_l$ is the compressive-complexity pooling of the $l$-th layer as defined in Section 3.2. We remark that MALCOM++ does not use the input preprocessing technique, so it does not need to compute the gradients for backpropagation, which eventually reduce its computational cost for the inference.

**Experimental settings.** For each method, the hyperparameters are adjusted by using 1000 in-distribution images and 1000 out-of-distribution images from the test dataset; this experimental setting is exactly the same with the setting in (Lee et al., 2018b). Specifically, for ODIN, we uses all the 2000 images to tune the hyperparameters $T_o \in \{1, 10, 100, 1000\}$ and $\epsilon_o \in \{0, 0.0005, 0.001, 0.0014, 0.002, 0.0024, 0.005, 0.01, 0.05, 0.1, 0.2\}$. In case of MAHALANOBIS, we split 2000 images into two groups; one of them is used to train the weights $\boldsymbol{\alpha}_m$ by logistic regression, and the other is for the validation of the input preprocessing hyperparameter $\epsilon_m \in \{0.0, 0.01, 0.005, 0.002, 0.0014, 0.001, 0.0005\}$. For fair comparisons, MALCOM++ uses the 1000 images to obtain weights $\boldsymbol{\alpha}_c$ which are the same with that for $\boldsymbol{\alpha}_m$. All experiments are repeated three times and the averages are reported.

**Results.** Table 4 shows the performances of out-of-distribution detection methods when out-of-distribution samples are used to find the optimal hyperparameter values. We observe that MALCOM++ shows slightly better performances than MAHALANOBIS when ResNet is used, even though MALCOM++ does not utilize the additional technique (i.e., input preprocessing). Our proposed method, MALCOM, does not use out-of-distribution samples for the validation, but it consistently outperforms ODIN that utilizes out-of-distibution samples; sometimes it even performs the best. Specifically, in case of CIFAR-10 for in-distribution and the cropped TinyImageNet for out-of-distribution, MALCOM achieves the best performance among all the other methods with DenseNet.

## C  VALIDATION ON ADVERSARIAL SAMPLES

In this section, we report the performance of out-of-distribution detection methods, validated by generated adversarial examples. The adversarial examples refer to the images that are hard to be distinguished with the original images by human eyes but misclassified by a classifier. Lee et al. (2018b) suggested to tune the hyperparameters by using generated adversarial examples rather than out-of-distribution samples. Although its generation process takes much cost, the validation on the adversarial examples generated from the training dataset also satisfies our constraints in Section 2.1.

**Adversarial examples.** We generate adversarial examples by using FGSM (Goodfellow et al., 2015). It generates adversarial examples by adding noise to the input images in the direction of the loss gradient. Given a loss function $\mathcal{L}(\boldsymbol{f}; y)$ and data with labels $(\boldsymbol{x}_i, y_i) \in \mathcal{D}_{train}$, the FGSM constructs adversarial examples as follows.

$$\tilde{\boldsymbol{x}}_i := \boldsymbol{x}_i + \epsilon_f \, \text{sign} \left( \nabla \mathcal{L}(\boldsymbol{f}(\boldsymbol{x}_i); y_i) \right), \tag{1}$$

where $\epsilon_f$ is the magnitude of noise. As in (Lee et al., 2018b), we set $\epsilon_f = 0.0525$ for DenseNet and $\epsilon_f = 0.0625$ for ResNet. The images that are still correctly classified after applying FGSM are excluded. The generated images are assumed to be out-of-distribution samples and the images from the training set are used for in-distribution samples.

**Experimental settings.** We use the same competing methods defined in Appendix B: ODIN (Liang et al., 2018) and MAHALANOBIS (Lee et al., 2018b). Since we do not assume any test distributions

Table 5: Performance of different out-of-distribution detection methods that utilize adversarial samples for the validation. For out-of-distribution datasets, 'C' and 'R' are the abbreviations for 'cropped' and 'resized', respectively. The best result is highlighted in boldface.

| ID | | OOD | TNR at TPR 95% | AUROC | Detection Acc. |
|---|---|---|---|---|---|
| | | | ODIN / Mahalanobis / MALCOM (ours) / MALCOM++(ours) | | |
| DenseNet | CIFAR-10 | SVHN | 38.68 / 82.66 / 94.03 / **95.65** | 88.71 / 96.55 / 98.56 / **98.93** | 82.72 / 91.02 / 94.54 / **95.43** |
| | | TinyIm(C) | 88.78 / 99.08 / **99.98** / 99.49 | 98.14 / 99.73 / **99.91** / 99.86 | 93.01 / 98.51 / **99.17** / 99.13 |
| | | TinyIm(R) | 85.20 / 92.23 / 93.90 / **95.59** | 97.44 / 98.38 / 98.61 / **99.05** | 91.75 / 94.32 / 94.49 / **95.38** |
| | | LSUN(C) | 78.19 / 97.76 / **99.87** / 99.16 | 96.58 / 99.38 / **99.83** / 99.77 | 90.84 / 97.36 / 98.70 / **98.72** |
| | | LSUN(R) | 91.20 / 95.47 / 95.43 / **96.71** | 98.53 / 98.89 / 98.95 / **99.26** | 93.87 / 95.48 / 95.25 / **96.09** |
| | | iSUN | 87.76 / 92.40 / 94.01 / **95.65** | 97.99 / 98.46 / 98.66 / **99.11** | 92.53 / 94.34 / 94.59 / **95.51** |
| | CIFAR-100 | SVHN | 39.49 / 66.31 / 63.31 / **82.65** | 88.21 / 92.48 / 93.09 / **95.47** | 80.72 / 85.48 / 86.32 / **89.73** |
| | | TinyIm(C) | 71.70 / 96.68 / **97.93** / 97.01 | 94.15 / 98.85 / 98.81 / **98.96** | 86.73 / 96.70 / 96.52 / **97.11** |
| | | TinyIm(R) | 43.14 / 81.30 / 85.79 / **88.65** | 85.34 / 96.39 / 97.12 / **97.57** | 77.19 / 90.74 / 91.33 / **92.23** |
| | | LSUN(C) | 80.10 / 94.61 / 94.96 / **95.95** | 96.31 / 98.11 / 98.30 / **98.51** | 89.82 / 95.26 / 95.49 / **96.28** |
| | | LSUN(R) | 41.54 / 86.75 / 88.43 / **91.70** | 85.71 / 97.04 / 97.59 / **97.94** | 77.36 / 92.05 / 92.68 / **93.48** |
| | | iSUN | 37.92 / 81.25 / 85.65 / **90.04** | 84.29 / 96.37 / 97.27 / **97.75** | 76.16 / 90.67 / 91.65 / **92.66** |
| | SVHN | CIFAR-10 | 69.14 / 96.02 / **96.04** / 94.99 | 91.83 / **98.78** / 98.63 / 98.51 | 86.54 / **96.00** / 95.62 / 95.23 |
| | | TinyIm(C) | 79.03 / **100.0 / 100.0 / 100.0** | 94.79 / **100.0 / 100.0 / 100.0** | 89.89 / 99.97 / **99.98** / 99.97 |
| | | TinyIm(R) | 79.73 / 99.50 / **99.96** / 99.44 | 94.76 / 99.86 / **99.87** / 99.85 | 90.21 / 98.89 / **98.94** / 98.83 |
| | | LSUN(C) | 76.71 / **100.0 / 100.0 / 100.0** | 93.76 / **100.0 / 100.0 / 100.0** | 88.76 / 99.97 / **99.98** / 99.97 |
| | | LSUN(R) | 77.07 / 99.59 / **100.0** / 99.49 | 94.07 / 99.88 / **99.89** / 99.85 | 89.16 / 99.18 / **99.23** / 99.07 |
| | | iSUN | 78.33 / 99.57 / **99.99** / 99.48 | 94.39 / **99.88 / 99.88** / 99.85 | 89.69 / 99.16 / **99.20** / 99.06 |
| ResNet | CIFAR-10 | SVHN | 34.04 / 73.54 / 80.54 / **94.47** | 86.17 / 93.92 / 96.80 / **98.84** | 78.43 / 86.94 / 91.37 / **94.79** |
| | | TinyIm(C) | 69.88 / 99.57 / 98.88 / **99.82** | 94.41 / 99.90 / 98.58 / **99.96** | 87.65 / 98.68 / 97.18 / **99.44** |
| | | TinyIm(R) | 67.43 / 90.98 / 89.62 / **96.74** | 93.68 / 98.27 / 98.25 / **99.31** | 86.52 / 93.52 / 93.12 / **96.03** |
| | | LSUN(C) | 66.28 / 98.93 / 98.04 / **99.71** | 93.58 / 99.79 / 99.48 / **99.94** | 86.73 / 97.88 / 96.69 / **99.22** |
| | | LSUN(R) | 68.63 / 94.91 / 91.56 / **97.84** | 93.77 / 98.87 / 98.51 / **99.48** | 86.70 / 95.03 / 93.75 / **96.75** |
| | | iSUN | 67.96 / 92.24 / 91.16 / **97.02** | 93.71 / 98.43 / 98.44 / **99.33** | 86.64 / 93.91 / 93.66 / **96.21** |
| | CIFAR-100 | SVHN | 11.63 / 41.49 / 63.22 / **82.49** | 72.48 / 81.01 / 93.01 / **96.29** | 68.44 / 73.38 / 85.02 / **89.95** |
| | | TinyIm(C) | 24.12 / 94.14 / 89.34 / **98.73** | 78.48 / 98.08 / 97.96 / **99.30** | 71.98 / 94.72 / 92.37 / **97.83** |
| | | TinyIm(R) | 32.45 / 56.17 / 64.04 / **80.46** | 82.25 / 91.28 / 92.44 / **96.26** | 74.24 / 84.70 / 83.80 / **90.86** |
| | | LSUN(C) | 11.47 / 88.15 / 89.53 / **98.23** | 64.61 / 97.07 / 97.98 / **99.38** | 62.51 / 91.86 / 92.44 / **97.22** |
| | | LSUN(R) | 30.00 / 46.47 / 60.57 / **75.78** | 80.57 / 88.07 / 91.51 / **95.13** | 73.02 / 81.76 / 82.60 / **89.31** |
| | | iSUN | 29.52 / 47.18 / 61.04 / **74.96** | 80.77 / 88.79 / 91.37 / **95.23** | 73.31 / 82.25 / 82.40 / **89.39** |
| | SVHN | CIFAR-10 | 78.28 / **97.80** / 90.42 / 97.21 | 92.95 / **99.38** / 98.02 / 99.29 | 90.05 / **97.21** / 94.46 / 96.73 |
| | | TinyIm(C) | 82.52 / 99.97 / **100.0** / 99.97 | 94.03 / **99.98** / 99.96 / **99.98** | 91.42 / 99.95 / 99.11 / **99.96** |
| | | TinyIm(R) | 79.05 / **99.70** / 97.69 / 99.69 | 93.53 / **99.90** / 99.23 / **99.90** | 90.42 / 99.08 / 96.39 / **99.09** |
| | | LSUN(C) | 79.63 / 99.95 / **100.0** / 99.96 | 93.06 / **99.98** / 99.90 / **99.98** | 90.53 / 99.94 / 98.57 / **99.95** |
| | | LSUN(R) | 74.45 / **99.72** / 97.13 / **99.72** | 91.53 / **99.91** / 99.06 / **99.91** | 88.94 / **99.29** / 96.23 / 99.27 |
| | | iSUN | 77.05 / **99.71** / 97.42 / **99.71** | 92.25 / **99.91** / 99.16 / **99.91** | 89.73 / **99.25** / 96.34 / 99.18 |

in this experiment, we use the whole training samples and adversarial examples for the validation. The only difference of the experimental setup from Appendix B is that we split the validation dataset into 80:20 for training the weights $\alpha_c$ (or $\alpha_m$) and determining the hyperparameter $\epsilon_m$, respectively. All experiments are repeated three times and the averages are reported.

**Results.** Table 5 shows the detection performances of MALCOM, MALCOM++ and the other competing methods when the generated adversarial samples are used for the validation. Our methods (i.e., MALCOM and MALCOM++) beat both ODIN and Mahalanobis in most cases, except for the case that SVHN is set to in-distribution. This result strongly indicates that the compressive-complexity pooling is still helpful and complements the limitation of the global average pooling, even in the situation where adversarial samples are available. In addition, we perform an ablation study to see how much each component affects the final detection performance. In order to control the effect of the input preprocessing, we exclude Mahalanobis and instead consider the method using the weighted sum of the scores computed based on global average pooling. As presented in Table 6, the method using all of the components consistently performs the best, similarly to the results of the previous ablation study (Table 2). From this observation, we conclude that

Table 6: Ablation results on analyzing the effect of each component: the weighted sum of the scores obtained from all layers (denoted by weighted sum), global average pooling, and compressive-complexity pooling. AUROC is used as an evaluation metric. The best results are marked in bold-face.

| Method | Weighted sum | Global average pooling | Compressive complexity pooling | CIFAR-10 | | CIFAR-100 | |
|---|---|---|---|---|---|---|---|
| | | | | TinyIm | LSUN | TinyIm | LSUN |
| BASELINE | - | - | - | 94.14 | 95.47 | 71.62 | 70.84 |
| MAHALANOBIS-vanilla | - | ✓ | - | 96.27 | 97.06 | 95.69 | 96.60 |
| MALCOM++ (ours) | - | - | ✓ | 93.26 | 94.42 | 92.29 | 92.28 |
| | - | ✓ | ✓ | 95.47 | 96.14 | 95.92 | 96.78 |
| | ✓ | ✓ | - | 98.83 | 99.17 | 97.43 | 97.91 |
| | ✓ | - | ✓ | 95.09 | 94.61 | 94.21 | 93.39 |
| | ✓ | ✓ | ✓ | **99.05** | **99.26** | **97.57** | **97.94** |

the compressive-complexity pooling is effective to detect out-of-distribution samples based on the Mahalanobis distance.

# D MORE RESULTS

Table 7 shows the detailed results for out-of-distribution detection. To perform further analysis, we test on the synthetic noise datasets which as in Liang et al. (2018).

Table 7: Performance of different out-of-distribution detection methods. The best result is highlighted in boldface for each setting.

| Model | ID | OOD | TNR at TPR 95% | AUROC | Detection Accuracy | AUPR(In) | AUPR(Out) |
|---|---|---|---|---|---|---|---|
| | | | BASELINE/ MAHALANOBIS-vanilla / MALCOM (ours) | | | | |
| DenseNet | CIFAR-10 | CIFAR-100 | **40.60(0.00)** / 18.04(0.00) / 19.53(0.71) | **89.30(0.00)** / 71.47(0.00) / 66.36(1.38) | **82.92(0.00)** / 65.56(0.00) / 61.97(0.75) | **89.37(0.00)** / 72.18(0.00) / 62.12(2.02) | **86.63(0.00)** / 69.62(0.00) / 67.55(0.98) |
| | | SVHN | 40.39(0.00) / 89.19(0.00) / **93.43(0.72)** | 89.88(0.00) / 97.64(0.00) / **98.37(0.22)** | 83.18(0.00) / 92.42(0.00) / **94.30(0.32)** | 83.73(0.00) / 94.54(0.00) / **95.87(1.11)** | 94.40(0.00) / 99.02(0.00) / **99.28(0.06)** |
| | | TinyIm(C) | 65.84(0.00) / 74.56(0.00) / **99.97(0.02)** | 95.37(0.00) / 95.39(0.00) / **99.91(0.01)** | 90.07(0.00) / 88.41(0.00) / **99.09(0.10)** | 96.47(0.00) / 95.76(0.00) / **99.92(0.01)** | 93.89(0.00) / 95.10(0.00) / **99.89(0.01)** |
| | | TinyIm(R) | 59.38(0.00) / 82.30(0.00) / **94.01(0.53)** | 94.14(0.00) / 96.27(0.00) / **98.52(0.28)** | 88.50(0.00) / 89.98(0.00) / **94.56(0.27)** | 95.32(0.00) / 96.15(0.00) / **97.98(0.79)** | 92.39(0.00) / 96.42(0.00) / **98.66(0.16)** |
| | | LSUN(C) | 61.38(0.00) / 52.91(0.00) / **99.85(0.08)** | 94.89(0.00) / 90.79(0.00) / **99.82(0.03)** | 89.83(0.00) / 82.77(0.00) / **98.59(0.21)** | 96.15(0.00) / 91.85(0.00) / **99.84(0.02)** | 93.24(0.00) / 89.57(0.00) / **99.79(0.03)** |
| | | LSUN(R) | 66.90(0.00) / 84.54(0.00) / **95.65(0.56)** | 95.47(0.00) / 97.06(0.00) / **98.96(0.09)** | 90.24(0.00) / 91.36(0.00) / **95.42(0.26)** | 96.51(0.00) / 97.24(0.00) / **98.92(0.20)** | 94.12(0.00) / 96.81(0.00) / **98.91(0.08)** |
| | | iSUN | 63.32(0.00) / 83.32(0.00) / **94.21(0.34)** | 94.84(0.00) / 96.59(0.00) / **98.70(0.07)** | 89.25(0.00) / 90.39(0.00) / **94.68(0.17)** | 96.35(0.00) / 96.92(0.00) / **98.52(0.09)** | 92.49(0.00) / 96.23(0.00) / **98.58(0.06)** |
| | | Uniform | 13.00(0.21) / **100.0(0.00)** / **100.0(0.00)** | 91.94(0.02) / **100.0(0.00)** / **100.0(0.00)** | 92.38(0.04) / **100.0(0.00)** / **100.0(0.00)** | 95.28(0.01) / **100.0(0.00)** / **100.0(0.00)** | 81.76(0.07) / **100.0(0.00)** / **100.0(0.00)** |
| | | Gaussian | 7.05(0.21) / **100.0(0.00)** / **100.0(0.00)** | 90.89(0.04) / **100.0(0.00)** / **100.0(0.00)** | 91.87(0.03) / **100.0(0.00)** / **100.0(0.00)** | 94.72(0.02) / **100.0(0.00)** / **100.0(0.00)** | 79.53(0.08) / **100.0(0.00)** / **100.0(0.00)** |
| | CIFAR-100 | CIFAR-10 | **18.50(0.00)** / 1.16(0.00) / 1.15(0.05) | **75.80(0.00)** / 49.86(0.00) / 45.21(0.19) | **69.70(0.00)** / 52.78(0.00) / 50.66(0.08) | **78.46(0.00)** / 53.99(0.00) / 48.90(0.08) | **71.87(0.00)** / 47.09(0.00) / 44.44(0.12) |
| | | SVHN | 26.24(0.00) / 45.01(0.00) / **64.85(2.22)** | 82.64(0.00) / 85.93(0.00) / **93.37(0.43)** | 75.54(0.00) / 78.61(0.00) / **86.60(0.56)** | 75.09(0.00) / 71.65(0.00) / **88.29(0.96)** | 90.36(0.00) / 92.73(0.00) / **96.10(0.25)** |
| | | TinyIm(C) | 32.74(0.00) / 52.85(0.00) / **98.10(0.15)** | 81.08(0.00) / 86.84(0.00) / **98.82(0.01)** | 72.88(0.00) / 79.05(0.00) / **96.64(0.09)** | 82.89(0.00) / 85.85(0.00) / **99.16(0.01)** | 80.00(0.00) / 87.28(0.00) / **98.16(0.02)** |
| | | TinyIm(R) | 17.26(0.00) / 81.99(0.00) / **85.27(0.18)** | 71.62(0.00) / 95.69(0.00) / **97.11(0.05)** | 65.71(0.00) / 89.53(0.00) / **91.24(0.12)** | 74.09(0.00) / 95.32(0.00) / **97.29(0.07)** | 68.95(0.00) / 95.87(0.00) / **96.95(0.05)** |
| | | LSUN(C) | 40.13(0.00) / 12.42(0.00) / **95.29(0.50)** | 85.44(0.00) / 60.59(0.00) / **98.35(0.06)** | 76.72(0.00) / 56.88(0.00) / **95.70(0.19)** | 86.98(0.00) / 61.70(0.00) / **98.83(0.05)** | 84.38(0.00) / 60.54(0.00) / **97.19(0.10)** |
| | | LSUN(R) | 16.44(0.00) / 84.49(0.00) / **87.20(0.55)** | 70.84(0.00) / 96.60(0.00) / **97.47(0.06)** | 65.01(0.00) / 91.03(0.00) / **92.42(0.12)** | 74.15(0.00) / 96.76(0.00) / **97.83(0.05)** | 67.87(0.00) / 96.12(0.00) / **96.88(0.06)** |
| | | iSUN | 14.81(0.00) / 82.13(0.00) / **84.79(0.53)** | 69.68(0.00) / 96.02(0.00) / **97.17(0.05)** | 63.95(0.00) / 89.77(0.00) / **91.42(0.14)** | 74.67(0.00) / 96.24(0.00) / **97.54(0.03)** | 63.98(0.00) / 95.55(0.00) / **96.54(0.07)** |
| | | Uniform | 0.13(0.03) / **100.0(0.00)** / **100.0(0.00)** | 54.69(0.14) / **100.0(0.00)** / **100.0(0.00)** | 65.90(0.07) / **100.0(0.00)** / **100.0(0.00)** | 69.17(0.09) / **100.0(0.00)** / **100.0(0.00)** | 47.04(0.08) / **100.0(0.00)** / **100.0(0.00)** |
| | | Gaussian | 0.00(0.00) / **100.0(0.00)** / **100.0(0.00)** | 36.84(0.12) / **100.0(0.00)** / **100.0(0.00)** | 59.96(0.07) / **100.0(0.00)** / **100.0(0.00)** | 57.00(0.09) / **100.0(0.00)** / **100.0(0.00)** | 39.46(0.04) / **100.0(0.00)** / **100.0(0.00)** |
| | SVHN | CIFAR-10 | 69.14(0.00) / 76.29(0.01) / **95.91(0.08)** | 91.83(0.00) / 96.38(0.00) / **98.60(0.01)** | 86.54(0.00) / 91.70(0.00) / **95.52(0.02)** | 95.37(0.00) / 98.73(0.00) / **99.52(0.00)** | 83.45(0.00) / 88.32(0.00) / **94.65(0.04)** |
| | | CIFAR-100 | 68.68(0.00) / 76.38(0.01) / **97.01(0.09)** | 91.38(0.00) / 96.36(0.00) / **98.88(0.01)** | 86.52(0.00) / 91.45(0.00) / **96.05(0.06)** | 94.83(0.00) / 98.70(0.00) / **99.61(0.00)** | 83.09(0.00) / 88.74(0.00) / **95.98(0.04)** |
| | | TinyIm(C) | 79.03(0.00) / 64.81(0.02) / **100.0(0.00)** | 94.79(0.00) / 95.03(0.00) / **100.0(0.00)** | 89.89(0.00) / 90.50(0.00) / **99.98(0.00)** | 97.19(0.00) / 98.26(0.00) / **100.0(0.00)** | 89.09(0.00) / 83.11(0.00) / **99.99(0.00)** |
| | | TinyIm(R) | 79.73(0.00) / 71.27(0.02) / **99.96(0.01)** | 94.76(0.00) / 96.03(0.00) / **99.87(0.00)** | 90.21(0.00) / 91.78(0.00) / **98.91(0.02)** | 97.04(0.00) / 98.63(0.00) / **99.95(0.00)** | 88.90(0.00) / 86.87(0.00) / **99.56(0.01)** |
| | | LSUN(C) | 76.71(0.00) / 70.58(0.01) / **100.0(0.00)** | 93.76(0.00) / 95.29(0.00) / **100.0(0.00)** | 88.76(0.00) / 89.74(0.00) / **99.98(0.00)** | 96.65(0.00) / 98.24(0.00) / **100.0(0.00)** | 87.42(0.00) / 86.76(0.00) / **99.99(0.00)** |
| | | LSUN(R) | 77.07(0.00) / 61.38(0.02) / **99.99(0.00)** | 94.07(0.00) / 95.07(0.00) / **99.89(0.00)** | 89.16(0.00) / 90.99(0.00) / **99.21(0.01)** | 96.70(0.00) / 98.33(0.00) / **99.96(0.00)** | 87.95(0.00) / 82.33(0.00) / **99.61(0.01)** |
| | | iSUN | 78.33(0.00) / 69.34(0.02) / **100.0(0.01)** | 94.39(0.00) / 95.86(0.00) / **99.88(0.00)** | 89.69(0.00) / 91.70(0.00) / **99.17(0.04)** | 97.17(0.00) / 98.70(0.00) / **99.96(0.00)** | 87.29(0.00) / 84.67(0.00) / **99.53(0.02)** |
| | | Uniform | 15.47(0.20) / 97.27(0.09) / **100.0(0.00)** | 67.78(0.18) / 98.91(0.01) / **100.0(0.00)** | 65.09(0.12) / 96.28(0.03) / **100.0(0.00)** | 62.28(0.23) / 99.17(0.01) / **100.0(0.00)** | 66.57(0.16) / 98.51(0.01) / **100.0(0.00)** |
| | | Gaussian | 34.91(0.51) / 78.77(0.25) / **100.0(0.00)** | 82.35(0.16) / 96.63(0.02) / **100.0(0.00)** | 76.49(0.10) / 93.92(0.06) / **100.0(0.00)** | 79.62(0.27) / 97.75(0.02) / **100.0(0.00)** | 80.67(0.18) / 93.66(0.05) / **100.0(0.00)** |
| ResNet | CIFAR-10 | CIFAR-100 | 33.13(0.00) / 40.26(0.00) / **46.23(0.33)** | 86.38(0.00) / 88.29(0.00) / **88.95(0.08)** | 80.30(0.00) / 81.78(0.00) / **81.96(0.03)** | 86.85(0.00) / 89.15(0.00) / **89.38(0.17)** | 83.44(0.00) / 85.67(0.00) / **87.19(0.05)** |
| | | SVHN | 32.19(0.00) / 54.52(0.00) / **79.19(0.58)** | 89.88(0.00) / 93.92(0.00) / **96.62(0.09)** | 85.06(0.00) / 89.13(0.00) / **91.13(0.12)** | 85.94(0.00) / 91.56(0.00) / **94.35(0.15)** | 93.57(0.00) / 95.95(0.00) / **98.27(0.06)** |
| | | TinyIm(C) | 49.11(0.00) / 66.12(0.00) / **99.03(0.10)** | 92.55(0.00) / 95.40(0.00) / **99.61(0.01)** | 86.85(0.00) / 90.38(0.00) / **97.25(0.06)** | 94.08(0.00) / 96.59(0.00) / **99.66(0.01)** | 90.05(0.00) / 93.27(0.00) / **99.56(0.01)** |
| | | TinyIm(R) | 44.05(0.00) / 69.28(0.00) / **89.65(0.12)** | 90.95(0.00) / 95.04(0.00) / **98.26(0.01)** | 84.95(0.00) / 88.59(0.00) / **93.13(0.04)** | 92.45(0.00) / 95.78(0.00) / **98.41(0.01)** | 88.27(0.00) / 94.04(0.00) / **98.18(0.01)** |
| | | LSUN(c) | 52.78(0.00) / 50.07(0.00) / **98.11(0.08)** | 93.30(0.00) / 94.29(0.00) / **99.49(0.01)** | 88.09(0.00) / 90.31(0.00) / **96.67(0.06)** | 94.78(0.00) / 96.04(0.00) / **99.55(0.01)** | 90.93(0.00) / 90.83(0.00) / **99.44(0.01)** |
| | | LSUN(r) | 45.14(0.00) / 78.27(0.00) / **91.55(0.21)** | 91.06(0.00) / 96.55(0.00) / **98.52(0.01)** | 85.26(0.00) / 90.68(0.00) / **93.77(0.02)** | 92.45(0.00) / 97.11(0.00) / **98.66(0.01)** | 88.59(0.00) / 95.81(0.00) / **98.42(0.01)** |
| | | iSUN | 44.49(0.00) / 76.84(0.00) / **91.32(0.29)** | 91.01(0.00) / 96.32(0.00) / **98.46(0.03)** | 85.01(0.00) / 90.36(0.00) / **93.67(0.07)** | 93.08(0.00) / 97.20(0.00) / **98.73(0.03)** | 87.29(0.00) / 95.05(0.00) / **98.21(0.04)** |
| | | Uniform | 48.91(0.29) / 99.99(0.01) / **100.0(0.00)** | 94.24(0.04) / 99.73(0.00) / **100.0(0.00)** | 91.04(0.09) / 98.75(0.02) / **100.0(0.00)** | 96.17(0.03) / 99.80(0.00) / **100.0(0.00)** | 89.77(0.07) / 99.50(0.01) / **100.0(0.00)** |
| | | Gaussian | 88.66(0.25) / **100.0(0.00)** / **100.0(0.00)** | 97.66(0.01) / 99.90(0.00) / **100.0(0.00)** | 95.01(0.07) / 99.54(0.01) / **100.0(0.00)** | 98.37(0.01) / 99.93(0.00) / **100.0(0.00)** | 96.13(0.02) / 99.79(0.00) / **100.0(0.00)** |
| | CIFAR-100 | CIFAR-10 | 18.54(0.00) / **18.96(0.00)** / 17.97(0.18) | **76.94(0.00)** / 76.82(0.00) / 76.36(0.05) | 70.91(0.00) / **71.64(0.00)** / 71.39(0.06) | **78.43(0.00)** / 75.87(0.00) / 75.85(0.09) | 72.53(0.00) / **73.06(0.00)** / 72.17(0.10) |
| | | SVHN | 19.88(0.00) / 43.20(0.00) / **61.36(0.95)** | 79.34(0.00) / 89.41(0.00) / **92.57(0.18)** | 73.20(0.00) / 81.99(0.00) / **84.44(0.20)** | 65.85(0.00) / 83.58(0.00) / **87.24(0.26)** | 88.27(0.00) / 94.28(0.00) / **96.41(0.09)** |
| | | TinyIm(C) | 16.87(0.00) / 34.65(0.00) / **88.77(0.61)** | 75.83(0.00) / 85.49(0.00) / **97.86(0.10)** | 70.14(0.00) / 78.18(0.00) / **92.09(0.25)** | 76.47(0.00) / 87.10(0.00) / **97.94(0.10)** | 71.24(0.00) / 82.64(0.00) / **97.90(0.10)** |
| | | TinyIm(R) | 20.18(0.00) / 20.28(0.00) / **64.20(0.77)** | 77.07(0.00) / 78.91(0.00) / **92.53(0.15)** | 70.76(0.00) / 72.03(0.00) / **83.86(0.17)** | 79.69(0.00) / 81.78(0.00) / **93.03(0.12)** | 73.16(0.00) / 74.44(0.00) / **92.53(0.18)** |
| | | LSUN(C) | 12.72(0.00) / 42.50(0.00) / **88.48(0.60)** | 70.46(0.00) / 87.44(0.00) / **97.77(0.13)** | 66.57(0.00) / 79.97(0.00) / **91.96(0.27)** | 68.11(0.00) / 88.42(0.00) / **97.84(0.12)** | 66.48(0.00) / 85.27(0.00) / **97.84(0.12)** |
| | | LSUN(R) | 18.39(0.00) / 19.62(0.00) / **60.64(1.20)** | 75.59(0.00) / 79.00(0.00) / **91.61(0.21)** | 69.75(0.00) / 72.28(0.00) / **82.69(0.22)** | 77.41(0.00) / 81.99(0.00) / **92.41(0.15)** | 71.71(0.00) / 74.10(0.00) / **91.33(0.27)** |
| | | iSUN | 16.56(0.00) / 18.07(0.00) / **61.18(0.40)** | 75.69(0.00) / 78.52(0.00) / **91.52(0.10)** | 70.11(0.00) / 72.19(0.00) / **82.56(0.13)** | 79.55(0.00) / 83.06(0.00) / **92.78(0.08)** | 68.76(0.00) / 71.07(0.00) / **90.74(0.12)** |
| | | Uniform | 0.69(0.08) / 67.38(0.36) / **100.0(0.00)** | 66.92(0.12) / 94.94(0.06) / **100.0(0.00)** | 72.40(0.09) / 88.47(0.12) / **100.0(0.00)** | 78.14(0.09) / 95.99(0.05) / **100.0(0.00)** | 54.71(0.09) / 93.34(0.06) / **100.0(0.00)** |
| | | Gaussian | 0.00(0.00) / 4.18(0.13) / **100.0(0.00)** | 38.35(0.12) / 81.22(0.11) / **100.0(0.00)** | 60.14(0.03) / 80.40(0.14) / **100.0(0.00)** | 58.24(0.11) / 87.66(0.09) / **100.0(0.00)** | 40.16(0.04) / 68.69(0.11) / **100.0(0.00)** |
| | SVHN | CIFAR-10 | 78.28(0.00) / 85.03(0.00) / **90.44(0.12)** | 92.95(0.00) / 97.04(0.00) / **98.01(0.02)** | 90.05(0.00) / 93.13(0.00) / **94.46(0.03)** | 94.77(0.00) / 98.98(0.00) / **99.31(0.01)** | 86.39(0.00) / 89.27(0.00) / **93.35(0.10)** |
| | | CIFAR-100 | 76.91(0.00) / 83.30(0.00) / **89.52(0.13)** | 92.52(0.00) / 96.80(0.00) / **97.96(0.02)** | 89.56(0.00) / 92.66(0.00) / **94.22(0.02)** | 94.50(0.00) / 98.87(0.00) / **99.29(0.01)** | 85.68(0.00) / 89.12(0.00) / **93.47(0.08)** |
| | | TinyIm(C) | 82.52(0.00) / 83.27(0.00) / **100.0(0.00)** | 94.03(0.00) / 96.88(0.00) / **99.96(0.00)** | 91.42(0.00) / 93.85(0.00) / **99.11(0.03)** | 95.27(0.00) / 98.98(0.00) / **99.98(0.00)** | 88.43(0.00) / 86.78(0.00) / **99.88(0.01)** |
| | | TinyIm(R) | 79.05(0.00) / 84.48(0.00) / **97.68(0.08)** | 93.53(0.00) / 96.95(0.00) / **99.22(0.02)** | 90.42(0.00) / 93.12(0.00) / **96.39(0.03)** | 95.41(0.00) / 98.94(0.00) / **99.72(0.01)** | 86.87(0.00) / 89.06(0.00) / **97.75(0.05)** |
| | | LSUN(C) | 79.63(0.00) / 84.07(0.00) / **100.0(0.00)** | 93.06(0.00) / 96.97(0.00) / **99.90(0.00)** | 90.53(0.00) / 93.24(0.00) / **98.54(0.05)** | 94.45(0.00) / 99.00(0.00) / **99.96(0.00)** | 87.26(0.00) / 87.77(0.00) / **99.73(0.01)** |
| | | LSUN(R) | 74.45(0.00) / 78.31(0.00) / **97.19(0.13)** | 91.53(0.00) / 96.15(0.00) / **99.04(0.02)** | 88.94(0.00) / 91.94(0.00) / **96.21(0.03)** | 93.76(0.00) / 98.65(0.00) / **99.66(0.01)** | 84.60(0.00) / 85.86(0.00) / **97.12(0.06)** |
| | | iSUN | 77.05(0.00) / 81.74(0.00) / **97.51(0.25)** | 92.25(0.00) / 96.60(0.00) / **99.18(0.04)** | 89.73(0.00) / 92.67(0.00) / **96.36(0.10)** | 94.58(0.00) / 98.92(0.00) / **99.73(0.01)** | 84.29(0.00) / 86.46(0.00) / **97.32(0.13)** |
| | | Uniform | 84.73(0.25) / 93.67(0.12) / **100.0(0.00)** | 95.35(0.05) / 98.05(0.01) / **100.0(0.00)** | 92.44(0.07) / 95.46(0.01) / **99.97(0.01)** | 92.23(0.11) / 98.64(0.00) / **100.0(0.00)** | 95.17(0.04) / 96.78(0.01) / **100.0(0.00)** |
| | | Gaussian | 85.76(0.16) / 92.04(0.08) / **100.0(0.00)** | 95.75(0.08) / 97.79(0.01) / **100.0(0.00)** | 92.82(0.07) / 95.18(0.02) / **100.0(0.00)** | 93.22(0.25) / 98.49(0.00) / **100.0(0.00)** | 95.46(0.05) / 96.12(0.01) / **100.0(0.00)** |

