# OpenReview forum: "Out-of-Distribution Image Detection Using the Normalized Compression Distance"
_ICLR.cc/2020/Conference — Reject_

### Official Review · AnonReviewer2 · 2019-10-22
**Official Blind Review #2**

**Rating:** 6

**Review:**

This paper proposes a new framework for out-of-distribution detection, based on global average pooling and spatial pattern of the feature maps to accurately identify out-of-distribution samples. MAHALANOBIS distance based methods were discussed, and the shortcomings of using mahalanobis distance were given (i.e. assumption that features are independent, etc). They propose to use compression based distance measures off the shelf from standard compression techniques to detect spatial feature patterns in feature space and demonstrate its effectiveness on several datasets and comparison with baselines is reported and well discussed.

The motivation of the proposed approach is clear, and the method seems novel. However, the experiments could have been done in a more complex setting. The out-of distribution samples pose a danger in safety critical applications such as autonomous driving, for example a car deployed in environment that it has not seen during the training might crash. So, it would be interesting to see the performance of both baselines and proposed approach in those settings where inputs are similar in nature but very different in some aspects. I do not necessarily see something wrong with the paper, but I'm not convinced of the significance (or sufficient efficiency) of the approach. There is also  theoretical guarantees showing exhaustiveness of the proposed methods in detecting all possible out-of distribution examples.


**Experience Assessment:**

I have published one or two papers in this area.

**Review Assessment: Checking Correctness Of Derivations And Theory:**

I assessed the sensibility of the derivations and theory.

**Review Assessment: Checking Correctness Of Experiments:**

I assessed the sensibility of the experiments.

**Review Assessment: Thoroughness In Paper Reading:**

I read the paper at least twice and used my best judgement in assessing the paper.

---

> ### Author Response · Authors · 2019-11-15
> **Response to Reviewer #2**
>
>  We are strongly convinced that out-of-distribution detection is important for AI safety, and we believe that there will be more studies on the task in the future. By the way, the settings you mentioned where the inputs are similar but very different in some aspects sound interesting. For now, the idea does not come to mind, but we think that it could be future work to propose the strict definition of “similar to A, but different from B”, formulate the settings, and conduct experiments. Last but not least, the intrinsic importance of our work is to make the out-of-distribution detection task more practical, which is slightly different in that most previous work just tried to improve the performance in existing settings. The compressive-complexity pooling, which we introduced by using the normalized compression distance (NCD), successfully improves the performance even with the constraints that makes the method much practical.

---

### Official Review · AnonReviewer1 · 2019-10-24
**Official Blind Review #1**

**Rating:** 3

**Review:**

After reading the other reviews and comments, I appreciate the effort by the Authors, but it looks like the paper still needs some work before being ready. So, I have decided to maintain my rating.

===================

The work proposes a system for detecting out-of-distribution images for neural networks under strict limitations of not retraining the network or tuning parameters with out of distribution validation data in mind using Compression Distance in a novel way.  The authors evaluate the method broadly and against the state of the art and provide a thorough explanation of the background material and formulation.

This work is strong in it’s cleverness, novelty, and evaluation when limited to the class of solution stipulated in section 2; however, it is not clear or presented why this restrictive choice is or could be necessary.    For this reason, I am borderline unless that caveat is addressed as described below, in which case I would be happy to accept.

It is unclear to me as it was not presented in the work when or if the problem that is being solved by the paper is particularly either important or frequent.  Obviously, not having to retrain is more efficient than having to and not having to validate on out-of-distribution samples is helpful in times when we don’t know them ahead of time, but it is unclear to me if it is the case that we will have a situation in which both of the above are, for instance, not possible at all.  To me, if the work could be changed to compare against works which are not so tightly constrained, not for the purposes of holding it to the same standard but to understand it’s relative standing, or to better justify the very strict constraints which somehow, despite out-of-distribution detection being a popular upcoming topic, apparently only has one other paper that matches it.

The paper could use some extra proofreading, for instance the first sentence of the abstract doesn’t make much grammatical sense especially with the first phrase included.
It may be nice to cite works such as http://citeseerx.ist.psu.edu/viewdoc/download?doi=10.1.1.132.6389&rep=rep1&type=pdf and others in that vein as this is certainly not the first work to involve compressive principles in image classification related tasks.

**Experience Assessment:**

I have read many papers in this area.

**Review Assessment: Checking Correctness Of Derivations And Theory:**

I carefully checked the derivations and theory.

**Review Assessment: Checking Correctness Of Experiments:**

I carefully checked the experiments.

**Review Assessment: Thoroughness In Paper Reading:**

I read the paper thoroughly.

---

> ### Author Response · Authors · 2019-11-15
> **Response to Reviewer #1**
>
> We really appreciate your constructive comments. We found them helpful and the followings are our responses to the major comments.
>
> - You mainly pointed out that our justification about both the constraints is not enough. In case of the first constraint, validating the models by using out-of-distribution test samples does not make sense, because the main motivation of out-of-distribution detection is that we are not able to know (and assume) the test distribution. For this reason, the usage of the best hyperparameter values found by a few out-of-distribution samples eventually becomes identical to assume a specific test distribution (including out-of-distribution samples), which makes the experiments unfair. In case of the second constraint, retraining the models to make it effective to detect out-of-distribution samples not only seems unnatural in that they are already being deployed, but also usually degrades the in-distribution classification performance [3]. Simply employing the pretrained deep neural network to compute the confidence score does not compromise the performance of in-distribution classification, and it is the main reason why this kind of approach has gained much attention.
>
> - In addition to the justification about the strict constraints above, we totally agree that our method needs to be compared against existing methods which are not so tightly constrained. Thus, we conducted the additional experiments of out-of-distribution detection in case that a few out-of-distribution samples (or adversarial samples) are available for validation (please refer to Appendix B and C). For competing methods, we considered 1) ODIN [2] and 2) Mahalanobis [1] that require the validation to determine the hyperparameter values. For fair comparisons, we additionally build a variant of our proposed method, termed as MALCOM++, which uses the weighted sum of the multiple scores for the final confidence score and the weights should be validated. As shown in Table 4 and 5, MALCOM++ consistently outperforms all the other methods in most cases, especially when validating with the adversarial samples. From this observation, we can conclude that the proposed compressive-complexity pooling is effective and helps to accurately identify the out-of-distribution images even without the strict constraint related to the validation.
>
> - We further polished the writing and corrected several typos. We agree with your comment suggesting to add missing related work about NCD on image classification, so added the citation in the paper.
>
> [1] Kimin Lee, Kibok Lee, Honglak Lee, and Jinwoo Shin.  A simple unified framework for detecting out-of-distribution samples and adversarial attacks.  NeurIPS 2018.
> [2] Shiyu Liang, Yixuan Li, and R. Srikant. Enhancing the reliability of out-of-distribution image detection in neural networks. In 6th International Conference on Learning Representations, ICLR 2018
> [3] Andrey Malinin and Mark Gales. Predictive uncertainty estimation via prior networks. NeurIPS 2018.

---

### Official Review · AnonReviewer3 · 2019-10-27
**Official Blind Review #3**

**Rating:** 3

**Review:**

** post rebuttal start **

After reading reviews and authors' response, I decided not to change my score.
I remark that this paper still requires a lot of revision; comparison in the main paper is somewhat unfair and all new results are in the appendix.
Also, the performance of their replication of the prior method is far lower than reported. In worst case, the performance gain from the compared method would be from their incorrect implementation on the prior works. In this kind of case, I suggest the authors to put {the numbers in the original paper} as well as {their replication} and claim that they fail to replicate the number. Ideally, if their method is evaluated in the same condition, it should outperform prior works in any case.


Detailed comments:

1-(a). Adversarial attack and OOD (which is hard to detect) are closely related: they are both in off-manifold. Their main difference would be, while adversarial attack is very close to the clean data in the data space, OOD is relatively far from the in-distribution in the data space.
Though it does not talk about OOD, you may refer to [R1] for analysis in perspective of manifold. The difficulty of OOD detection can be considered to be coming from overlapped manifolds in the latent space.

[R1] Stutz et al. Disentangling Adversarial Robustness and Generalization. In CVPR, 2019.


1-(b). Though the numbers are much lower than those reported in the original paper, I am happy to see the fair comparison. However, according to the original paper, simple FGSM is used for validation, so I am not sure such a huge difference can actually happen. In this kind of case, I suggest the authors to put {the numbers in the original paper} as well as {their replication} and claim that they fail to replicate the number.


2. I am happy to see that their revised method has better performance.

** post rebuttal end **


- Summary:
This paper proposes an out-of-distribution detection (OOD) method under constraints that 1) no OOD is available for validation and 2) model parameters should be unchanged. They specifically address a problem of the state-of-the-art method satisfying the constraints, and propose a new distance metric inspired by data compression. Experimental results on several benchmarks with different deep neural network architectures support their claim.


- Decision and supporting arguments:
Weak reject.

1. The problem setting is clear and their approach is interesting and makes sense. However, the method for comparison is not properly set. As the authors addressed, Mahalanobis detector proposed by Lee et al. (2018b) requires validation to determine weights for feature ensembling, but the validation can be done without OOD data by generating adversarial samples as proposed in the same paper. Although Table 2 in Lee et al. (2018b) shows that the performance is not better than the case when we have an explicit OOD data for validation, it reasonably works well. Therefore, rather than comparing with the vanilla version (only using last latent space) or the alternative "assemble" method (concatenating all average-pooled features), they had to compare their method with the model validated by adversarial samples, which essentially satisfies the constraints.

2. Also, I wonder the main body of the proposed method itself is really effective or some minor tweak they made is essential. According to Table 2 in the submission, their method is better than "Mahalanobis vanilla" only when all components are applied. Though the idea is interesting, I am skeptical about the effectiveness of the proposed method.


- Comments:
1. As addressed by the authors, feature concatenation ("assemble") is not effective for the Mahalanobis method but the proposed method. How about to do an ablation study about weighted averaging vs. concatenation on the proposed method as well? Again, weights can be validated by adversarial samples to satisfy the constraints.

**Experience Assessment:**

I have published in this field for several years.

**Review Assessment: Checking Correctness Of Derivations And Theory:**

I assessed the sensibility of the derivations and theory.

**Review Assessment: Checking Correctness Of Experiments:**

I assessed the sensibility of the experiments.

**Review Assessment: Thoroughness In Paper Reading:**

I read the paper at least twice and used my best judgement in assessing the paper.

---

> ### Author Response · Authors · 2019-11-15
> **Response to Reviewer #3 (1)**
>
> We really appreciate your constructive comments. We found them helpful and the followings are our responses to the major comments.
>
> 1-(a). First of all, we are not sure that utilizing generated adversarial examples for validation does make sense due to the intrinsic difference between the adversarial examples and out-of-distribution samples that we want to detect, and it could be an another research topic by itself. In case that we use the adversarial examples in place of the out-of-distribution samples for the validation, we have to carefully determine “what makes samples like out-of-distribution”. Also, the adversarial examples looks similar with their original images to our eyes, so it is not consistent with the basic assumption of out-of-distribution data, which might differ in some respect from the training (or in-distribution) data. Although Lee et al. [1] tried to validate the model by using generated adversarial examples as you mentioned, we wonder if the results would be robust with respect to how the adversarial examples are constructed. In this sense, we think that constructing such adversarial examples for out-of-distribution data detection is one of interesting research topics but slightly different from the problem that we want to tackle.
>
> 1-(b). Nevertheless, it is true that adjusting hyperparmeters by utilizing adversarial examples does not violate our constraints as you mentioned. Thus, we conducted the additional experiments about this, and please refer to Table 5 in Appendix C. We could not figure out the details about how Lee et al. [1] generated the adversarial examples from the training set, but we tried to reproduce the experimental setting as much as possible. We observe that our method, MALCOM, shows the comparable performance to the Mahalanobis detector except for a few cases (e.g. AUROC 81.01% -> 93.01% for (id, ood)=(CIFAR-10, SVHN) using ResNet). Furthermore, as adversarial samples are available for validation, we extended our method to use the weighted summation of confidence scores from multiple layers similarly to Mahalanobis, and we named this method as MALCOM++. With the help of our proposed compressive-complexity pooling, MALCOM++ outperforms Mahalanobis in most cases.

---

> > ### Author Response · Authors · 2019-11-15
> > **Response to Reviewer #3 (2)**
> >
> > 2. As our method aims to additionally capture the spatial information of feature maps, it works well when both the global average pooling and the compressive-complexity pooling are used. We can explain this observation for both the cases 1) without the concatenation and 2) with the concatenation:
> >
> > 2-(a). Without the concatenation:
> > The feature vector obtained by the global average pooling on the last layer of a CNN model is very effective to characterize input images for classification. This is because the CNN model is already trained for classification based on the global average pooling layer, which feeds the processed data into the last linear layer. Thus, the CNN model focuses on training the feature vector obtained by global average pooling of the last convolutional layer. Therefore, the pooled feature vector already has rich information related to its target class, so that it can describe the image [2]. This is why the vanilla version of the Mahalanobis method works so well and our compressive-complexity pooling sometimes fails to extract other information.
> >
> > 2-(b). With the concatenation:
> > Except for the last convolutional layer, there is no guarantee that the global average pooling can capture much information related to the class. In fact, Table 1 shows that AUPR(In) of Mahalanobis-assemble is significantly low, which means that out-of-distribution samples are more recognized as in-distribution samples when we concatenate feature vectors average-pooled from hidden layers than when we do not. This indicates that the global average pooling on feature maps of lower layers cannot correctly capture the class information, because the spatial information becomes more important to capture low-level features of classes. For this reason, our compressive-complexity pooling that utilizes the spatial information of feature maps could be effective to discriminate out-of-distribution feature maps from in-distribution feature maps especially in lower layer, and it eventually helps to detect out-of-distribution samples.
> >
> >
> > In summary, the performance of MALCOM could be comparable to the method using the global average pooling when only the last hidden layer is used as discussed in 2-(a). However, we think that the important part is the observation from 2-(b) in that MALCOM achieves the best performance by effectively capturing the spatial information in lower feature maps, which the existing global average pooling is not able to do.
> >
> >
> > * As you suggested, we added the ablation study about weighted averaging on the proposed method (please refer to Appendix C). We first build a variant of our proposed method, termed as MALCOM++, which uses the weighted sum of the multiple scores for the final confidence score and the weights are validated by adversarial samples. We observe that the weighted sum (utilizing the adversarial samples for the validation) performs better than the concatenation (not using adversarial samples at all) by the help of the adversarial samples in general. However, as presented in Table 6, MALCOM++ shows the best performance, which strongly indicates that the proposed compressive-complexity pooling is consistently effective regardless of whether the adversarial examples are used or not.
> >
> > [1] Kimin Lee, Kibok Lee, Honglak Lee, and Jinwoo Shin.  A simple unified framework for detectingout-of-distribution samples and adversarial attacks.  NeurIPS 2018
> > [2] Babenko, Artem, and Victor Lempitsky. Aggregating deep convolutional features for image retrieval. arXiv 2015

---

### Author Response · Authors · 2019-11-15
**Overview of revision**

Dear reviews,
Considering the constructive comments from reviewers, we revised the paper as follows:


- We added Figure 1 to help you better understand the overall process of our compressive-complexity pooling and its difference from the global average pooling.

- We added Appendix B with Table 4, which is about the experiments in case that out-of-distribution samples are used for validation.

- We added Appendix C with Table 5 & 6, which is about the experiments in case that generated adversarial samples are used for validation.

- The performances of Mahalanobis-vanilla on the experiment (in-distribution:”CIFAR-100”, ResNet) have been wrongly reported and we corrected it.

- We corrected several typos.

(+) For reproducibility, we upload the code although it needs to be refactored.
(https://github.com/malcom2020/malcom)

---

### Decision · Program_Chairs · 2019-12-19

**Decision:**

Reject

**Comment:**

This paper proposes an out-of-distribution detection (OOD) method without assuming OOD in validation.

As reviewers mentioned, I think the idea is interesting and the proposed method has potential. However, I think the paper can be much improved and is not ready to publish due to the followings given reviewers' comments:

(a) The prior work also has some experiments without OOD in validation, i.e., use adversarial examples (AE) instead in validation. Hence, the main motivation of this paper becomes weak unless the authors justify enough why AE is dangerous to use in validation.

(b) The performance of their replication of the prior method is far lower than reported. I understand that sometimes it is not easy to reproduce the prior results. In this case, one can put the numbers in the original paper. Or, one can provide detailed analysis why the prior method should fail in some cases.

(c) The authors follow exactly same experimental settings in the prior works. But, the reported score of the prior method is already very high in the settings, and the gain can be marginal. Namely, the considered settings are more or less "easy problems". Hence, additional harder interesting OOD settings, e.g., motivated by autonomous driving, would strength the paper.

Hence, I recommend rejection.